# Adversarial Bandits Against Arbitrary Strategies

**Jung-hun Kim**                                                                      *junghun.kim@ensae.fr*
*CREST, ENSAE, IP Paris*
*FairPlay joint team, France*

**Se-Young Yun**                                                                      *yunseyoung@kaist.ac.kr*
*KAIST AI, South Korea*

Reviewed on OpenReview: *https://openreview.net/forum?id=x4QrOh8uCs*

## Abstract

We study the adversarial bandit problem against arbitrary strategies, where the difficulty is captured by an unknown parameter $S$, which is the number of switches in the best arm in hindsight. To handle this problem, we adopt the master-base framework using the online mirror descent method (OMD). We first provide a master-base algorithm with simple OMD, achieving $\tilde{O}(S^{1/2}K^{1/3}T^{2/3})$, in which $T^{2/3}$ comes from the variance of loss estimators. To mitigate the impact of the variance, we propose using adaptive learning rates for OMD and achieve $\tilde{O}(\min\{\sqrt{SKT\rho}, S\sqrt{KT}\})$, where $\rho$ is a variance term for loss estimators.

## 1 Introduction

The bandit problem is a fundamental framework in sequential decision-making that addresses the exploration-exploitation trade-off. At each time step, an agent selects an action (referred to as an arm) and observes a corresponding reward or loss. In applications such as recommendation systems, arms may represent items presented to users, and user preferences can evolve over time. This dynamic nature can be modeled by allowing the identity of the best arm to change, leading to the notion of switching best arms.

To capture such evolving environments, it is natural to consider performance against a changing sequence of optimal arms rather than a single fixed one. The problem of competing against switching arms has been extensively studied. In the expert setting with full-information feedback (Cesa-Bianchi et al., 1997), several algorithms (Daniely et al., 2015; Jun et al., 2017) have been developed that achieve a near-optimal regret bound of $\tilde{O}(\sqrt{ST})$ for $S$-switch regret (formally defined later), without requiring prior knowledge of the number of switches $S$. In contrast, the bandit setting presents a greater challenge, as the agent only observes the feedback for the selected arm rather than the full loss vector, making the problem significantly harder than in the full-information case.

In the stochastic bandit setting where each arm's reward distribution may change over time, commonly referred to as the non-stationary bandit problem, several studies have addressed the challenge of switching environments (Garivier & Moulines, 2008; Auer et al., 2019; Russac et al., 2019; Suk & Kpotufe, 2022). Notably, Auer et al. (2019) and Suk & Kpotufe (2022) achieved near-optimal regret bounds of $\tilde{O}(\sqrt{SKT})$ without requiring prior knowledge of the number of switches $S$. However, these methods rely on stochastic assumptions and are not applicable in the adversarial setting, where losses can be chosen arbitrarily. For the adversarial bandit setting, `EXP3.S` algorithm (Auer et al., 2002) achieves a regret bound of $\tilde{O}(\sqrt{SKT})$, but assumes that $S$ is known in advance. When $S$ is unknown, the Bandit-over-Bandit (`BOB`) approach has been proposed, achieving a regret bound of $\tilde{O}(\sqrt{SKT} + T^{3/4})$ (Cheung et al., 2019; Foster et al., 2020). More recently, Luo et al. (2022) studied the problem of switching adversarial linear bandits and achieved a regret of $\tilde{O}(\sqrt{dST})$ under the assumption that $S$ is known.

In this paper, we study adversarial bandit problems against arbitrarily switching arms. Crucially, we allow the number of switches $S$ to be unknown to the agent, thereby targeting arbitrary strategies without prior

knowledge of their complexity. To address this setting, we adopt the master-base framework combined with the online mirror descent (OMD) method, inspired by Agarwal et al. (2017); Pacchiano et al. (2020); Luo et al. (2022). We begin by analyzing a master-base algorithm that employs OMD with a negative entropy regularizer, and show that it achieves a regret bound of $\tilde{O}(S^{1/2}K^{1/3}T^{2/3})$. However, this approach relies on a fixed learning rate, which limits its ability to adapt to the variance of the loss estimators, leading to a suboptimal regret term proportional to $T^{2/3}$.

Building on this analysis, we propose using adaptive learning rates within the OMD framework to better control the variance of loss estimators. This refinement leads to an improved regret bound of $\tilde{O}(\min\{\sqrt{SKT\rho}, S\sqrt{KT}\})$ with respect to $T$, where $\rho$ captures the variance associated with a comparator strategy. Crucially, rather than employing the standard negative entropy regularizer, we adopt a log-barrier regularizer, which enables tighter control over worst-case scenarios in terms of the variance term $\rho$.

## 2 Problem Statement

We now formalize the problem setting. Let $\mathcal{A} = [K]$ denote the set of $K$ arms, and let $\boldsymbol{l}_t \in [0,1]^K$ be the loss vector at time $t$, where $l_t(a)$ denotes the loss incurred by arm $a \in [K]$ at time $t$. The environment is adversarial, generating an arbitrary sequence of loss vectors $\boldsymbol{l}_1, \boldsymbol{l}_2, \ldots, \boldsymbol{l}_T \in [0,1]^K$ over a time horizon of $T$ rounds. At each round $t \in [T]$, the agent selects an arm $a_t \in [K]$ and observes only the loss $l_t(a_t)$ of the chosen arm. Our goal is to minimize the $S$-switch regret, which measures the performance gap between the agent and the best sequence of actions that switches arms at most $S$ times.

Formally, let $\kappa = \{\kappa_1, \kappa_2, \ldots, \kappa_T\} \in [K]^T$ denote a sequence of comparator actions. For a positive integer $S < T$, define the set of sequences with at most $S$ switches as

$$B_S = \left\{ \kappa \in [K]^T : \sum_{t=1}^{T-1} \mathbb{1}\{\kappa_t \neq \kappa_{t+1}\} \leq S \right\}.$$

Then, the $S$-switch regret is defined by

$$R_S(T) = \max_{\kappa \in B_S} \sum_{t=1}^{T} \mathbb{E}[\boldsymbol{l}_t(a_t)] - \boldsymbol{l}_t(\kappa_t).$$

We consider the setting where the number of switches $S$ is unknown to the agent (i.e., not provided in advance). Our goal is to design algorithms that perform well against any sequence of actions, without relying on prior knowledge of $S$. This requires the development of universal algorithms that achieve tight regret bounds uniformly over all values of $S \in [T-1]$, where $S$ characterizes the hardness of the problem (Auer et al., 2002). Notably, this setting generalizes non-stationary stochastic bandit problems in which the switching parameter is unknown (Auer et al., 2019; Chen et al., 2019).

## 3 Algorithms and Regret Analysis

To address the problem of adversarial bandits with an unknown switching budget $S$, we propose algorithms based on the master-base framework combined with the online mirror descent (OMD) method.

### 3.1 Master-Base Framework

In the master-base framework, a master algorithm selects one among several base algorithms at each round, and the selected base then chooses an arm to play. Since the true switch parameter $S$ is not known in advance, we instantiate each base algorithm with a different candidate value of $S$ from a predefined set.

Let $\mathcal{H}$ denote the set of candidate values for $S$, defined as:

$$\mathcal{H} = \{T^0, T^{\frac{1}{\lceil \log T \rceil}}, T^{\frac{2}{\lceil \log T \rceil}}, \ldots, T\}.$$

Each base is associated with a candidate $h \in \mathcal{H}$ and tunes its learning rate accordingly. Let $H = |\mathcal{H}| = O(\log T)$ be the number of base algorithms. When there is no ambiguity, we refer to a base instantiated with parameter $h \in \mathcal{H}$ simply as base $h$. Define $h^\dagger \in \mathcal{H}$ as the largest candidate not exceeding the true (unknown) value of $S$: $h^\dagger = \max\{h \in \mathcal{H} : h \leq S\}$. By construction, this ensures

$$e^{-1} S \leq h^\dagger \leq S,$$

which guarantees that $h^\dagger$ provides a near-optimal approximation of $S$.

## 3.2 Online Mirror Descent (OMD)

We now present the *Online Mirror Descent* (OMD) method (Lattimore & Szepesvári, 2020), which serves as the fundamental update rule for the master and each base algorithm in our framework. OMD generalizes classic online gradient descent by incorporating a flexible geometry through a regularizer.

Let $F : \mathbb{R}^d \to \mathbb{R}$ be a convex and differentiable regularizer function. This regularizer defines the *Bregman divergence* between two points $\boldsymbol{p}, \boldsymbol{q} \in \mathbb{R}^d$:

$$D_F(\boldsymbol{p}, \boldsymbol{q}) = F(\boldsymbol{p}) - F(\boldsymbol{q}) - \langle \nabla F(\boldsymbol{q}), \boldsymbol{p} - \boldsymbol{q} \rangle.$$

In our context, the decision variable is a probability distribution $\boldsymbol{p}_t$ over $d$ arms, i.e., $\boldsymbol{p}_t \in \mathcal{P}_d$ where $\mathcal{P}_d$ denotes the $d$-dimensional probability simplex. At each round $t$, given a loss vector $\boldsymbol{l} \in \mathbb{R}^d$, OMD computes the next distribution by solving the following optimization problem:

$$\boldsymbol{p}_{t+1} = \arg\min_{\boldsymbol{p} \in \mathcal{P}_d} \left\{ \langle \boldsymbol{p}, \boldsymbol{l} \rangle + D_F(\boldsymbol{p}, \boldsymbol{p}_t) \right\}. \tag{1}$$

This formulation reflects the trade-off between exploiting current loss information (via the linear term) and staying close to the previous distribution (via the Bregman divergence).

In practice, the update of equation 1 is often implemented in two steps for computational convenience:

$$\tilde{\boldsymbol{p}}_{t+1} = \arg\min_{\boldsymbol{p} \in \mathbb{R}^d} \left\{ \langle \boldsymbol{p}, \boldsymbol{l} \rangle + D_F(\boldsymbol{p}, \boldsymbol{p}_t) \right\},$$

$$\boldsymbol{p}_{t+1} = \arg\min_{\boldsymbol{p} \in \mathcal{P}_d} D_F(\boldsymbol{p}, \tilde{\boldsymbol{p}}_{t+1}).$$

Here, the first step performs an unconstrained update in the dual space defined by $F$, while the second step projects the intermediate point back onto the simplex, ensuring that the updated distribution is a valid probability vector. The choice of regularizer $F$ determines the specific algorithm instance; it typically includes a learning rate parameter that controls the step size and will be specified in subsequent sections.

Finally, in the bandit setting, since the learner does not observe the full loss vector $\boldsymbol{l}_t$ but only the loss of the selected arm, the true loss must be replaced with an appropriate unbiased estimator. This ensures that OMD remains applicable even with partial feedback.

## 3.3 Master-Base OMD

We employ an OMD framework with a hierarchical structure consisting of a master and multiple bases. We first present a simple Master-Base OMD algorithm (Algorithm 1) that employs the negative entropy regularizer:

$$F_\eta(\boldsymbol{p}) = (1/\eta) \sum_{i=1}^d (p(i) \log p(i) - p(i)),$$

where $\boldsymbol{p} \in \mathbb{R}^d$, $p(i)$ denotes the $i$-index entry for $\boldsymbol{p}$, and $\eta$ is a learning rate. This regularizer is commonly used in adversarial bandit algorithms, including the well-known `EXP3` algorithm (Auer et al., 2002).

In Algorithm 1, at each round $t$, the master algorithm selects a base $h_t \in \mathcal{H}$ according to a probability distribution $\boldsymbol{p}_t$. The selected base $h_t$ then chooses an arm $a_t \in [K]$ using its internal distribution $\boldsymbol{p}_{t,h_t}$

and observes the incurred loss $l_t(a_t)$. From this partial feedback, we construct unbiased estimators: $l'_t(h)$ estimates the loss for each base $h \in \mathcal{H}$, and $l''_{t,h}(a)$ estimates the loss for each arm $a \in [K]$ under base $h$. Using these estimators, the algorithm updates both the master distribution $\boldsymbol{p}_{t+1}$ and each base's distribution $\boldsymbol{p}_{t+1,h}$ via OMD.

To control variance in the estimator $l'_t(h) = l_t(a_t)\mathbb{1}(h = h_t)/p_t(h)$, we define the master's update domain as a clipped probability simplex $\mathcal{P}_H^\alpha = \mathcal{P}_H \cap [\alpha, 1]^H$ for some $\alpha > 0$. This clipping ensures that each base is selected with non-negligible probability, preventing high variance in the importance-weighted loss estimates. Within this domain, the update of $\boldsymbol{p}_{t+1}$ is then computed using the negative entropy regularizer with learning rate $\eta$.

Each base $h \in \mathcal{H}$ maintains its own arm selection distribution $\boldsymbol{p}_{t,h}$. The update for this distribution is also based on the negative entropy regularizer but with a learning rate that adapts to the candidate switch parameter $h$:

$$\xi(h) = \frac{h^{1/2}}{K^{1/3}T^{2/3}}.$$

This choice of $\xi(h)$ helps base $h$ adapt to environments with up to $h$ switches in the best arm sequence. The domain for updating $\boldsymbol{p}_{t,h}$ is $\mathcal{P}_K^\beta = \mathcal{P}_K \cap [\beta, 1]^K$ for $\beta > 0$. Unlike $\alpha$, which controls variance in master-level estimation, $\beta$ acts as a regularization mechanism to stabilize learning under the switching best arms in hindsight.

---

**Algorithm 1** Master-base OMD

---

**Input:** $T$, $K$, $\mathcal{H}$.
**Initialization:** $\alpha = K^{1/3}/(T^{1/3}H^{1/2})$, $\beta = 1/(KT)$, $\eta = 1/\sqrt{TH}$, $\xi(h) = h^{1/2}/(K^{1/3}T^{2/3})$, $p_1(h) = 1/H$, $p_{1,h}(a) = 1/K$ for $h \in \mathcal{H}$ and $a \in [K]$.
**for** $t = 1, \ldots, T$ **do**
    **Select a base and an arm**:
    Draw $h_t \sim$ probabilities $\{p_t(h)\}_{h \in \mathcal{H}}$.
    Draw $a_{t,h_t} \sim$ probabilities $\{p_{t,h_t}(a)\}_{a \in [K]}$.
    Pull $a_t = a_{t,h_t}$ and Receive $l_t(a_{t,h_t}) \in [0, 1]$.
    **Obtain loss estimators**:
    $l'_t(h_t) = \frac{l_t(a_{t,h_t})}{p_t(h_t)}$ and $l'_t(h) = 0$ for $h \in \mathcal{H}/\{h_t\}$.
    $l''_{t,h_t}(a_{t,h_t}) = \frac{l'_t(h_t)}{p_{t,h_t}(a_{t,h_t})}$ and $l''_{t,h}(a) = 0$ for $h \in \mathcal{H}/\{h_t\}$, $a \in [K]/\{a_{t,h_t}\}$.
    **Update distributions**:
    $\boldsymbol{p}_{t+1} = \arg\min_{\boldsymbol{p} \in \mathcal{P}_H^\alpha} \langle \boldsymbol{p}, \boldsymbol{l}'_t \rangle + D_{F_\eta}(\boldsymbol{p}, \boldsymbol{p}_t)$
    $\boldsymbol{p}_{t+1,h} = \arg\min_{\boldsymbol{p} \in \mathcal{P}_K^\beta} \langle \boldsymbol{p}, \boldsymbol{l}''_{t,h} \rangle + D_{F_{\xi(h)}}(\boldsymbol{p}, \boldsymbol{p}_{t,h})$ for all $h \in \mathcal{H}$
**end for**

---

Now we provide a regret bound for the algorithm in the following theorem.

**Theorem 3.1.** *For any switch number $S \in [T-1]$, Algorithm 1 achieves a regret bound of*

$$R_S(T) = \tilde{O}(S^{1/2}K^{1/3}T^{2/3})$$

*Proof Sketch.* Here, we provide a proof sketch, and the full version is provided in Appendix A.1.

In our proof, we decompose the regret into two parts: one is the regret from the master selecting a base at each time, and the other is the regret from the base selecting an arm at each time. Let $t_s$ be the time when the $s$-th switch of the best arm happens and $t_{S+1} - 1 = T$, $t_0 = 1$. Also let $t_{s+1} - t_s = T_s$. For any $t_s$ for

all $s \in [0, S]$, the $S$-switch regret can be expressed as

$$R_S(T) = \sum_{t=1}^{T} \mathbb{E}\left[l_t(a_t)\right] - \sum_{s=0}^{S} \min_{k_s \in [K]} \sum_{t=t_s}^{t_{s+1}-1} l_t(k_s)$$

$$= \sum_{t=1}^{T} \mathbb{E}\left[l_t(a_{t,h_t})\right] - \sum_{t=1}^{T} \mathbb{E}\left[l_t(a_{t,h^\dagger})\right] + \sum_{t=1}^{T} \mathbb{E}\left[l_t(a_{t,h^\dagger})\right] - \sum_{s=0}^{S} \min_{k_s \in [K]} \sum_{t=t_s}^{t_{s+1}-1} l_t(k_s), \tag{2}$$

in which the first two terms are closely related with the regret from the master algorithm against the near optimal base $h^\dagger$, and the remaining terms are related with the regret from $h^\dagger$ base algorithm against the best arms in hindsight. We note that the algorithm does not need to know $h^\dagger$ in prior and $h^\dagger$ is brought here only for regret analysis.

**Regret from the near-optimal base.** First we provide a bound for the following regret from base $h^\dagger$:

$$\sum_{t=1}^{T} \mathbb{E}\left[l_t(a_{t,h^\dagger})\right] - \sum_{s=0}^{S} \min_{k_s \in [K]} \sum_{t=t_s}^{t_{s+1}-1} l_t(k_s),$$

where the first term is the loss from the base and the second one is the loss from the optimal arm in hindsight. Let $k_s^* = \arg\min_{k \in [K]} \sum_{t=t_s}^{t_{s+1}-1} l_t(k)$ and $\boldsymbol{e}_{j,K}$ denote the unit vector with 1 at $j$-index and 0 at the rest of $K-1$ indices. Then, we have

$$\sum_{t=t_s}^{t_{s+1}-1} \mathbb{E}\left[l_t(a_{t,h^\dagger}) - l_t(k_s^*)\right] \leq \beta T_s(K-1) + \max_{\boldsymbol{p} \in \mathcal{P}_K^\beta} \mathbb{E}\left[\sum_{t=t_s}^{t_{s+1}-1} \langle \boldsymbol{p}_{t,h^\dagger} - \boldsymbol{p}, \boldsymbol{l}''_{t,h^\dagger} \rangle\right], \tag{3}$$

where the first term in the last inequality is obtained from the clipped domain $\mathcal{P}_K^\beta$ and the second term is obtained from the unbiased estimator $\boldsymbol{l}''_{t,h^\dagger}$ such that $\mathbb{E}[\boldsymbol{l}''_{t,h^\dagger}|\mathcal{F}_{t-1}] = \mathbb{E}[\boldsymbol{l}_t|\mathcal{F}_{t-1}]$ where $\mathcal{F}_{t-1}$ denotes the natural filtration generated by the history up to round $t-1$. We can observe that the clipped domain controls the distance between the initial distribution at $t_s$ and the best arm unit vector for the time steps over $[t_s, t_{s+1} - 1]$. Let

$$\tilde{\boldsymbol{p}}_{t+1,h^\dagger} = \arg\min_{\boldsymbol{p} \in \mathbb{R}^K} \langle \boldsymbol{p}, \boldsymbol{l}''_{t,h^\dagger} \rangle + D_{F_{\xi(h^\dagger)}}(\boldsymbol{p}, \boldsymbol{p}_{t,h^\dagger}).$$

Then, by solving the optimization problem, for all $k \in [K]$ we can obtain

$$\tilde{p}_{t+1,h^\dagger}(k) = p_{t,h^\dagger}(k) \exp(-\xi(h^\dagger)l''_{t,h^\dagger}(k)).$$

For the second term of the last inequality in equation 3, we have for any $\boldsymbol{p} \in \mathcal{P}_K^\beta$,

$$\sum_{t=t_s}^{t_{s+1}-1} \langle \boldsymbol{p}_{t,h^\dagger} - \boldsymbol{p}, \boldsymbol{l}''_{t,h^\dagger} \rangle \leq D_{F_{\xi(h^\dagger)}}(\boldsymbol{p}, \boldsymbol{p}_{t_s,h^\dagger}) + \sum_{t=t_s}^{t_{s+1}-1} D_{F_{\xi(h^\dagger)}}(\boldsymbol{p}_{t,h^\dagger}, \tilde{\boldsymbol{p}}_{t+1,h^\dagger}). \tag{4}$$

The first term is for the initial point diameter at time $t_s$ and the second term is for the divergence of the updated policy. Using the definition of the Bregman divergence and the fact that $p_{t_s,h^\dagger}(k) \geq \beta$, the initial point diameter term can be shown to be bounded as follows:

$$D_{F_{\xi(h^\dagger)}}(\boldsymbol{p}, \boldsymbol{p}_{t_s,h^\dagger}) \leq \frac{\log(1/\beta)}{\xi(h^\dagger)}. \tag{5}$$

Next, for the updated policy divergence term, using $\tilde{p}_{t+1,h^\dagger}(k) = p_{t,h^\dagger}(k) \exp(-\xi(h^\dagger)l''_{t,h^\dagger}(k))$ for all $k \in [K]$, we have

$$\sum_{t=t_s}^{t_{s+1}-1} \mathbb{E}\left[D_{F_{\xi(h^\dagger)}}(\boldsymbol{p}_{t,h^\dagger}, \tilde{\boldsymbol{p}}_{t+1,h^\dagger})\right] \leq \frac{\xi(h^\dagger)KT_s}{2\alpha}. \tag{6}$$

Then from equation 3, equation 4, equation 5, and equation 6, by summing up over $s \in [S]$, we have

$$\sum_{t=1}^{T} \mathbb{E}\left[l_t(a_{t,h^\dagger})\right] - \sum_{s=0}^{S} \min_{k_s \in [K]} \sum_{t=t_s}^{t_{s+1}-1} l_t(k_s) \leq \beta T(K-1) + \frac{S\log(1/\beta)}{\xi(h^\dagger)} + \frac{\xi(h^\dagger)KT}{2\alpha}. \tag{7}$$

Next, we provide a bound for the following regret from the master:

$$\sum_{t=1}^{T} \mathbb{E}\left[l_t(a_{t,h_t})\right] - \sum_{t=1}^{T} \mathbb{E}\left[l_t(a_{t,h^\dagger})\right].$$

Let $\tilde{\boldsymbol{p}}_{t+1} = \arg\min_{\boldsymbol{p} \in \mathbb{R}^H} \langle \boldsymbol{p}, \boldsymbol{l}'_t \rangle + D_{F_\eta}(\boldsymbol{p}, \boldsymbol{p}_t)$ and $\boldsymbol{e}_{h,H}$ denote the unit vector with 1 at base $h$-index and 0 at the rest of $H-1$ indices. For ease of presentation, we define $\tilde{l}_t(h) = l_t(a_{t,h})$. Then, we have

$$\sum_{t=1}^{T} \mathbb{E}\left[l_t(a_{t,h_t}) - l_t(a_{t,h^\dagger})\right] \leq \alpha T(H-1) + \max_{\boldsymbol{p} \in \mathcal{P}_H^\alpha} \mathbb{E}\left[\sum_{t=1}^{T} \langle \boldsymbol{p}_t - \boldsymbol{p}, \tilde{\boldsymbol{l}}_t \rangle\right]. \tag{8}$$

**Regret from the master.** For bounding the second term in equation 8, which arises from the master, we use the following: for any $\boldsymbol{p} \in \mathcal{P}_H^\alpha$

$$\sum_{t=1}^{T} \langle \boldsymbol{p}_t - \boldsymbol{p}, \tilde{\boldsymbol{l}}_t \rangle \leq F_\eta(\boldsymbol{p}) - F_\eta(\boldsymbol{p}_1) + \sum_{t=1}^{T} D_{F_\eta}(\boldsymbol{p}_t, \tilde{\boldsymbol{p}}_{t+1}). \tag{9}$$

From equation 8 and equation 9, we have

$$\sum_{t=1}^{T} \mathbb{E}\left[l_t(a_{t,h_t})\right] - \sum_{t=1}^{T} \mathbb{E}\left[l_t(a_{t,h^\dagger})\right]$$

$$\leq \alpha T(H-1) + \max_{\boldsymbol{p} \in \mathcal{P}_H^\alpha} \mathbb{E}\left[F_\eta(\boldsymbol{p}) - F_\eta(\boldsymbol{p}_1) + \sum_{t=1}^{T} D_{F_\eta}(\boldsymbol{p}_t, \tilde{\boldsymbol{p}}_{t+1})\right]$$

$$\leq \alpha T(H-1) + \frac{\log(H)}{\eta} + \frac{\eta T H}{2}. \tag{10}$$

**Overall Regret.** Therefore, putting equation 2, equation 7, and equation 10 altogether, we have

$$R_S(T) = \sum_{t=1}^{T} \mathbb{E}\left[l_t(a_t)\right] - \sum_{s=0}^{S} \min_{1 \leq k_s \leq K} \sum_{t=T_s}^{T_{s+1}-1} l_t(k_s)$$

$$\leq \alpha T H + \frac{\log(H)}{\eta} + \frac{\eta T H}{2} + \beta T(K-1) + \frac{S\log(1/\beta)}{\xi(h^\dagger)} + \frac{\xi(h^\dagger)KT}{2\alpha}$$

$$= \tilde{O}(S^{1/2}T^{2/3}K^{1/3}),$$

where $\alpha = K^{1/3}/(T^{1/3}H^{1/2})$, $\beta = 1/(KT)$, $\eta = 1/\sqrt{TH}$, $\xi(h^\dagger) = (h^\dagger)^{1/2}/(K^{1/3}T^{2/3})$, $h^\dagger = \Theta(S)$, and $H = \log(T)$. This completes the proof. $\square$

Compared to the prior parameter-free algorithm based on the Bandit-over-Bandit (`BOB`) approach (Cheung et al., 2019), which incurs a suboptimal dependence on the time horizon $T$, specifically, a regret term of order $T^{3/4}$, our algorithm achieves a tighter regret bound in terms of $T$. In particular, when $T = \omega(S^6 K^4)$, Algorithm 1 achieves a tighter regret bound than that of `BOB`.

However, the regret bound achieved by Algorithm 1 remains of order $O(T^{2/3})$, rather than the optimal $O(\sqrt{T})$, due to the high variance in the loss estimators. This variance arises from the double sampling process—first selecting a base, then an arm—at each round. To address this issue, we next propose an improved algorithm that leverages *adaptive* learning rates to better control the variance of the estimators.

### 3.4 Master-Base OMD with Adaptive Learning Rates

We now propose Algorithm 2, which incorporates adaptive learning rates to better control the variance of the loss estimators. We begin by describing the base algorithm. Each base employs the negative entropy regularizer, but with a time-varying adaptive learning rate $\xi_t(h)$, defined as:

$$F_{\xi_t(h)}(\boldsymbol{p}) = \frac{1}{\xi_t(h)} \sum_{i=1}^{d} (p(i) \log p(i) - p(i)),$$

where $\boldsymbol{p} \in \mathbb{R}^d$ is a probability distribution over arms. The learning rate $\xi_t(h)$ is dynamically adjusted at each round $t$ based on the variance of the loss estimators, and is given by:

$$\xi_t(h) = \sqrt{h/(KT\rho_t(h))},$$

where $\rho_t(h)$ is a variance threshold term that will be defined later. This formulation ensures that when the variance of the estimators is small, a larger learning rate is used—allowing for more aggressive updates—while high variance naturally leads to more conservative updates. Notably, this adaptive base algorithm is effectively combined with the master employing log-barrier regularization to control the regret due to variance, resulting in equation 15, which introduces a novel integration of adaptive learning and log-barrier regularization for variance control.

For the master algorithm, inspired by the approach in Agarwal et al. (2017), we employ a log-barrier regularizer with increasing learning rates. This design introduces a negative bias term that effectively offsets the variance arising from the base algorithms, particularly by addressing the worst-case scenario in terms of the variance threshold $\rho_t(h^\dagger)$. The log-barrier regularizer is defined as:

$$G_{\boldsymbol{\eta}_t}(\boldsymbol{p}) = - \sum_{i=1}^{d} \frac{\log p(i)}{\eta_t(i)},$$

where $\boldsymbol{p} \in \mathcal{P}_d$ is the master distribution, and $\boldsymbol{\eta}_t = (\eta_t(1), \dots, \eta_t(d))$ denotes the vector of learning rates at time $t$. We describe the learning rate update procedures for both the master and the base algorithms in Algorithm 2; all other components remain identical to those in Algorithm 1. The variance of the loss estimator $l'_{t+1}(h) = l_t(a_{t+1,h})\mathbf{1}(h_{t+1} = h)/p_{t+1}(h)$ for base $h$ is given by $1/p_{t+1}(h)$. If this variance exceeds the threshold $\rho_t(h)$, i.e., $1/p_{t+1}(h) > \rho_t(h)$, the master increases the learning rate as: $\eta_{t+1}(h) = \gamma\eta_t(h)$ for some fixed $\gamma > 1$. Simultaneously, the variance threshold is updated to: $\rho_{t+1}(h) = 2/p_{t+1}(h)$ which is also used to adaptively tune the base learning rate $\xi_t(h)$ as described earlier. If the variance does not exceed the threshold, both $\eta_t(h)$ and $\rho_t(h)$ remain unchanged from the previous time step.

In the following theorem, we provide a regret bound of Algorithm 2.

**Theorem 3.2.** *For any switch number $S \in [T-1]$ and any $\rho \geq \mathbb{E}[\rho_T(h^\dagger)]$, Algorithm 2 achieves a regret bound of*

$$R_S(T) = \tilde{O}\left(\min\left\{\sqrt{SKT\rho}, S\sqrt{KT}\right\}\right).$$

*Proof Sketch.* Here, we provide a proof sketch, and the full version is provided in Appendix A.2. As in the Theorem 3.1, we decompose the regret into two parts: one is the regret from the master selecting a base at each time, and the other is the regret from the base selecting an arm at each time. Let $t_s$ be the time when the $s$-th switch of the best arm happens and $t_{S+1} - 1 = T$, $t_0 = 1$. Also let $t_{s+1} - t_s = T_s$. For any $t_s$ for all $s \in [0, S]$, the $S$-switch regret can be expressed as

$$R_S(T) = \sum_{t=1}^{T} \mathbb{E}\left[l_t(a_t)\right] - \sum_{s=0}^{S} \min_{k_s \in [K]} \sum_{t=t_s}^{t_{s+1}-1} l_t(k_s)$$

$$= \sum_{t=1}^{T} \mathbb{E}\left[l_t(a_{t,h_t})\right] - \sum_{t=1}^{T} \mathbb{E}\left[l_t(a_{t,h^\dagger})\right] + \sum_{t=1}^{T} \mathbb{E}\left[l_t(a_{t,h^\dagger})\right] - \sum_{s=0}^{S} \min_{k_s \in [K]} \sum_{t=t_s}^{t_{s+1}-1} l_t(k_s). \tag{11}$$

---

**Algorithm 2** Master-base OMD with adaptive learning rates

---

**Input:** $T$, $K$, $\mathcal{H}$

**Initialization:** $\alpha = 1/(TH)$, $\beta = 1/(TK)$, $\gamma = e^{\frac{1}{\log T}}$, $\eta = \sqrt{H/T}$, $\rho_1(h) = 2H$, $\eta_1(h) = \eta$, $p_1(h) = 1/H$, $p_{1,h}(a) = 1/K$ for $h \in \mathcal{H}$ and $a \in [K]$.

**for** $t = 1, \ldots, T$ **do**

    **Select a base and an arm:**

    Draw $h_t \sim$ probabilities $\{p_t(h)\}_{h \in \mathcal{H}}$.

    Draw $a_{t,h_t} \sim$ probabilities $\{p_{t,h_t}(a)\}_{a \in [K]}$.

    Pull $a_t = a_{t,h_t}$ and Receive $l_t(a_{t,h_t}) \in [0,1]$.

    **Update loss estimators:**

    $l'_t(h_t) = \frac{l_t(a_{t,h_t})}{p_t(h_t)}$ and $l'_t(h) = 0$ for $h \in \mathcal{H}/\{h_t\}$.

    $l''_{t,h_t}(a_{t,h_t}) = \frac{l'_t(h_t)}{p_{t,h_t}(a_{t,h_t})}$ and $l''_{t,h}(a) = 0$ for $h \in \mathcal{H}/\{h_t\}$, $a \in [K]/\{a_{t,h_t}\}$.

    **Update distributions:**

    $\boldsymbol{p}_{t+1} = \arg\min_{\boldsymbol{p} \in \mathcal{P}_H^\alpha} \langle \boldsymbol{p}, \boldsymbol{l}'_t \rangle + D_{G_{\eta_t}}(\boldsymbol{p}, \boldsymbol{p}_t)$

    $\boldsymbol{p}_{t+1,h} = \arg\min_{\boldsymbol{p} \in \mathcal{P}_K^\beta} \langle \boldsymbol{p}, \boldsymbol{l}''_{t,h} \rangle + D_{F_{\xi_t(h)}}(\boldsymbol{p}, \boldsymbol{p}_{t,h})$ for $h \in \mathcal{H}$

    **Update learning rates:**

    For $h \in \mathcal{H}$

      If $\frac{1}{p_{t+1}(h)} > \rho_t(h)$, then

        $\rho_{t+1}(h) = \frac{2}{p_{t+1}(h)}, \eta_{t+1}(h) = \gamma \eta_t(h)$.

      Else, $\rho_{t+1}(h) = \rho_t(h), \eta_{t+1}(h) = \eta_t(h)$.

**end for**

---

**Regret from the near-optimal base.** First we provide a bound for the following regret from base $h^\dagger$. From equation 3, we can obtain

$$\sum_{t=t_s}^{t_{s+1}-1} \mathbb{E}\left[l_t(a_{t,h^\dagger})\right] - \sum_{t=t_s}^{t_{s+1}-1} l_t(k_s) \leq \beta T_s K + \mathbb{E}\left[\max_{p \in \mathcal{P}_K^\beta} \sum_{t=t_s}^{t_{s+1}-1} \langle \boldsymbol{p}_{t,h^\dagger} - \boldsymbol{p}, \boldsymbol{l}''_{t,h^\dagger} \rangle\right]. \tag{12}$$

Then for the second term of the last inequality in equation 12, we provide the following lemma.

**Lemma 3.3.** *For any $\boldsymbol{p} \in \mathcal{P}_K^\beta$ we can show that*

$$\sum_{t=t_s}^{t_{s+1}-1} \mathbb{E}\left[\langle \boldsymbol{p}_{t,h^\dagger} - \boldsymbol{p}, \boldsymbol{l}''_{t,h^\dagger} \rangle\right] \leq \mathbb{E}\left[2\log(1/\beta)\sqrt{\frac{KT\rho_T(h^\dagger)}{h^\dagger}} + \frac{T_s}{2}\sqrt{\frac{SK\rho_T(h^\dagger)}{T}}\right].$$

Then from equation 8 and Lemma 3.3, we have

$$\sum_{t=1}^{T} \mathbb{E}\left[l_t(a_{t,h^\dagger})\right] - \sum_{s=0}^{S} \min_{1 \leq k_s \leq K} \sum_{t=T_s}^{T_{s+1}-1} l_t(k_s) \leq \beta T(K-1) + \mathbb{E}\left[2S\log(1/\beta)\sqrt{\frac{KT\rho_T(h^\dagger)}{h^\dagger}} + \frac{1}{2}\sqrt{TSK\rho_T(h^\dagger)}\right]. \tag{13}$$

**Regret from the master.** Next, we provide a bound for the regret from the master in the following:

$$\sum_{t=1}^{T} \mathbb{E}\left[l_t(a_{t,h_t})\right] - \sum_{t=1}^{T} \mathbb{E}\left[l_t(a_{t,h^\dagger})\right] \leq O\left(\frac{H\log(T)}{\eta} + T\eta\right) - \mathbb{E}\left[\frac{\rho_T(h^\dagger)}{40\eta\log T}\right] + \alpha T(H-1). \tag{14}$$

The negative bias term in equation 14 is derived from the log-barrier regularizer and increasing learning rates $\eta_t(h)$. This term is critical to bound the worst case regret which will be shown soon. Also, $H\log(T)/\eta$ is obtained from $H\log(1/(H\alpha))/\eta$ considering the clipped domain.

**Overall Regret.** Then, putting equation 11, equation 13 and equation 14 altogether, we have

$$R_S(T) = \sum_{t=1}^{T} \mathbb{E}\left[l_t(a_t)\right] - \sum_{s=0}^{S} \min_{1 \le k_s \le K} \sum_{t=T_s}^{T_{s+1}-1} l_t(k_s) = \tilde{O}\left(\mathbb{E}\left[\sqrt{SKT\rho_T(h^\dagger)}\right]\right) - \mathbb{E}\left[\frac{\rho_T(h^\dagger)\sqrt{TK}}{40\sqrt{H}\log(T)}\right], \quad (15)$$

Then, we can obtain

$$R_S(T) = \tilde{O}\left(\min\left\{\sqrt{SKT\rho}, S\sqrt{KT}\right\}\right),$$

where $\tilde{O}(S\sqrt{KT})$ is obtained from the worst case of $\rho_T(h^\dagger)$. The worst case can be found by considering a maximum value of the concave bound of the last equality in equation 15 with variable $\rho_T(h^\dagger) > 0$ such that $\rho_T(h^\dagger) = \tilde{\Theta}(S)$. This concludes the proof. $\square$

We now provide a comparison of regret bounds with an existing approach. our regret bound depends on $\rho$, which captures the variance of the loss estimators $l_t'(h^\dagger)$ over the time horizon $t \in [T]$. While the bound is variance-dependent, it is noteworthy that in the worst case, it is always upper bounded by $\tilde{O}(S\sqrt{KT})$. Importantly, Algorithm 2 achieves a tight dependence of $O(\sqrt{T})$ in the regret bound, in contrast to Algorithm 1 and BOB (Cheung et al., 2019; Foster et al., 2020), which incur suboptimal $T^{2/3}$ and $T^{3/4}$ term, respectively. Therefore, when $T$ is sufficiently large, Algorithm 2 yields a strictly better regret guarantee than Algorithm 1 and BOB. We also note that the variance term $\rho$ is commonly observed in Luo et al. (2022), but we control the worst case without information of $S$ using an adaptive learning rate.

**Remark 3.4.** *Since we incorporate $\rho_t(h)$ into the adaptive learning rate for each base $h$ as $\xi_t(h) = \sqrt{h/KT\rho_t(h)}$, we can optimize the regret bound to depend on $\sqrt{\rho_T(h^\dagger)}$, as demonstrated in Lemma 3.3. Notably, this adaptive base algorithm is effectively combined with the master employing log-barrier regularization Agarwal et al. (2017) to control the regret due to variance, resulting $\sqrt{S\rho_T(h^\dagger)} - \rho_T(h^\dagger)$. This integration is the main reason why Algorithm 2 can achieve a order of $\sqrt{T}$, even in the worst case, without the knowledge of $S$.*

**Remark 3.5.** *Our algorithms are designed to perform well across different regimes, particularly with respect to $T$ and $S$. To recall the regret bounds, Algorithm 1 achieves a regret of $\tilde{O}(S^{1/2}K^{1/3}T^{2/3})$, while Algorithm 2 achieves $\tilde{O}(\min\{\sqrt{SKT\rho}, S\sqrt{KT}\})$. This indicates that Algorithm 1 is advantageous when $S$ is large, whereas Algorithm 2 is preferable for larger $T$ due to its use of an adaptive learning rate that accounts for the variance of the loss estimator.*

**Remark 3.6** (Implementation)**.** *For implementation of our algorithms, we briefly describe how to update the policy $\boldsymbol{p}_t$ using OMD. Let $\widehat{l}_s(a)$ denote a loss estimator for $l_s(a)$ for action $a \in [d]$ and $s \in [T]$. For the negative-entropy regularizer, by solving the optimization in equation 1 (see Lattimore & Szepesvári (2020)), we obtain*

$$p_t(a) = \frac{\exp\left(-\eta \sum_{s=1}^{t-1} \widehat{l}_s(a)\right)}{\sum_{b\in[d]} \exp\left(-\eta \sum_{s=1}^{t-1} \widehat{l}_s(b)\right)}.$$

*For the log-barrier regularizer, the solution is $p_t(a) = (\eta \sum_{s=1}^{t-1} \widehat{l}_s(a) + Z)^{-1}$, where $Z$ is the normalization constant ensuring $\sum_a p_t(a) = 1$ (Luo et al., 2022). When the feasible set is the clipped simplex $\mathcal{P}_d^{\epsilon/d} = \{p \in \Delta_d : p(a) \ge \epsilon/d\}$ for $0 < \epsilon < 1$, a computationally simple way to enforce feasibility is to mix with the uniform distribution:*

$$\bar{p}_t(a) \;\leftarrow\; (1-\epsilon)\,p_t(a) + \epsilon/d \quad \text{for all } a \in [d].$$

*This "uniform-mixing" trick, following Auer et al. (2002), guarantees $\bar{p}_t \in \mathcal{P}_d^{\epsilon/d}$ to ensure a bounded variance of the loss estimator. We emphasize that this is an implementation convenience rather than the exact Bregman projection onto $\mathcal{P}_d^{\epsilon/d}$; a rigorous regret analysis under this specific implementation is left for future work.*

**Remark 3.7.** *The regret bounds in Theorems 3.1 and 3.2 also extend to non-stationary stochastic bandit problems with unknown switching parameters, where the reward distributions may change over time. This generalization is possible because the adversarial bandit setting encompasses the stochastic setting as a special case.*

## 4 Conclusion

In this paper, we studied the adversarial bandit problem with $S$-switch regret, where the agent competes against any sequence of arms that switches at most $S$ times, without prior knowledge of $S$. To address this challenge, we proposed two algorithms based on the master-base framework integrated with the Online Mirror Descent (OMD) method.

First, we introduced Algorithm 1, which employs a simple OMD update with a fixed learning rate and achieves a regret bound of $\tilde{O}(S^{1/2}K^{1/3}T^{2/3})$. To further improve performance with respect to $T$, we proposed Algorithm 2, which incorporates adaptive learning rates to control the variance of the loss estimators. This leads to an improved regret bound of $\tilde{O}(\min\{\sqrt{SKT\rho}, S\sqrt{KT}\})$ where $\rho$ captures the variance of the estimators associated with the near-optimal base.

## Acknowledgements

We thank the Action Editor and the reviewers for their constructive and insightful feedback. J. Kim acknowledges the support of ANR through the PEPR IA FOUNDRY project (ANR-23-PEIA-0003) and the Doom project (ANR-23-CE23-0002), as well as the ERC through the Ocean project (ERC-2022-SYG-OCEAN-101071601).

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

## A  Appendix

### A.1  Proof of Theorem 3.1

Let $t_s$ be the time when the $s$-th switch of the best arm happens and $t_{S+1} - 1 = T$, $t_0 = 1$. Also let $t_{s+1} - t_s = T_s$. For any $t_s$ for all $s \in [0, S]$, the $S$-switch regret can be expressed as

$$
R_S(T) = \sum_{t=1}^{T} \mathbb{E}\left[l_t(a_t)\right] - \sum_{s=0}^{S} \min_{k_s \in [K]} \sum_{t=t_s}^{t_{s+1}-1} l_t(k_s)
$$

$$
= \sum_{t=1}^{T} \mathbb{E}\left[l_t(a_{t,h_t})\right] - \sum_{t=1}^{T} \mathbb{E}\left[l_t(a_{t,h^\dagger})\right] + \sum_{t=1}^{T} \mathbb{E}\left[l_t(a_{t,h^\dagger})\right] - \sum_{s=0}^{S} \min_{k_s \in [K]} \sum_{t=t_s}^{t_{s+1}-1} l_t(k_s), \qquad (16)
$$

in which the first two terms are closely related with the regret from the master algorithm against the near optimal base $h^\dagger$, and the remaining terms are related with the regret from $h^\dagger$ base algorithm against the best arms in hindsight. We note that the algorithm does not need to know $h^\dagger$ in prior and $h^\dagger$ is brought here only for regret analysis.

First we provide a bound for the following regret from base $h^\dagger$:

$$
\sum_{t=1}^{T} \mathbb{E}\left[l_t(a_{t,h^\dagger})\right] - \sum_{s=0}^{S} \min_{k_s \in [K]} \sum_{t=t_s}^{t_{s+1}-1} l_t(k_s).
$$

Let $k_s^* = \arg\min_{k \in [K]} \sum_{t=t_s}^{t_{s+1}-1} l_t(k)$ and $e_{j,K}$ denote the unit vector with 1 at $j$-index and 0 at the rest of $K - 1$ indices. Then, we have

$$
\sum_{t=t_s}^{t_{s+1}-1} \mathbb{E}\left[l_t(a_{t,h^\dagger}) - l_t(k_s^*)\right]
$$

$$
= \sum_{t=t_s}^{t_{s+1}-1} \mathbb{E}\left[\langle \boldsymbol{p}_{t,h^\dagger} - \boldsymbol{e}_{k_s^*,K}, \boldsymbol{l}_t \rangle\right]
$$

$$
\leq \max_{\boldsymbol{p} \in \mathcal{P}_K^\beta} \mathbb{E}\left[\sum_{t=t_s}^{t_{s+1}-1} \langle \boldsymbol{p} - \boldsymbol{e}_{k_s^*,K}, \boldsymbol{l}_t \rangle + \sum_{t=t_s}^{t_{s+1}-1} \langle \boldsymbol{p}_{t,h^\dagger} - \boldsymbol{p}, \boldsymbol{l}_t \rangle\right]
$$

$$
\leq \beta T_s(K-1) + \max_{\boldsymbol{p} \in \mathcal{P}_K^\beta} \mathbb{E}\left[\sum_{t=t_s}^{t_{s+1}-1} \langle \boldsymbol{p}_{t,h^\dagger} - \boldsymbol{p}, \boldsymbol{l}''_{t,h^\dagger} \rangle\right], \qquad (17)
$$

where the first term in the last inequality is obtained from the clipped domain $\mathcal{P}_K^\beta$ and the second term is obtained from the unbiased estimator $\boldsymbol{l}''_{t,h^\dagger}$ such that $\mathbb{E}[\boldsymbol{l}''_{t,h^\dagger}|\mathcal{F}_{t-1}] = \mathbb{E}[\boldsymbol{l}_t|\mathcal{F}_{t-1}]$ where $\mathcal{F}_{t-1}$ denotes the natural filtration generated by the history up to round $t - 1$. We can observe that the clipped domain

controls the distance between the initial distribution at $t_s$ and the best arm unit vector for the time steps over $[t_s, t_{s+1} - 1]$. Let

$$\tilde{\boldsymbol{p}}_{t+1,h^\dagger} = \underset{\boldsymbol{p} \in \mathbb{R}^K}{\arg\min} \langle \boldsymbol{p}, \boldsymbol{l}''_{t,h^\dagger} \rangle + D_{F_{\xi(h^\dagger)}}(\boldsymbol{p}, \boldsymbol{p}_{t,h^\dagger}).$$

Then, by solving the optimization problem, for all $k \in [K]$ we can obtain

$$\tilde{p}_{t+1,h^\dagger}(k) = p_{t,h^\dagger}(k) \exp(-\xi(h^\dagger) l''_{t,h^\dagger}(k)).$$

For the second term of the last inequality in equation 17, we provide a lemma in the following.

**Lemma A.1** (Theorem 28.4 and Eq. 28.11 in Lattimore & Szepesvári (2020)). *For any $\boldsymbol{p} \in \mathcal{P}_K^\beta$ we have*

$$\sum_{t=t_s}^{t_{s+1}-1} \langle \boldsymbol{p}_{t,h^\dagger} - \boldsymbol{p}, \boldsymbol{l}''_{t,h^\dagger} \rangle \leq D_{F_{\xi(h^\dagger)}}(\boldsymbol{p}, \boldsymbol{p}_{t_s,h^\dagger}) + \sum_{t=t_s}^{t_{s+1}-1} D_{F_{\xi(h^\dagger)}}(\boldsymbol{p}_{t,h^\dagger}, \tilde{\boldsymbol{p}}_{t+1,h^\dagger}).$$

*Proof.* For completeness, we provide a proof for this lemma. Fix any $\boldsymbol{p} \in \mathcal{P}_K^\beta$. By the first-order optimality condition of the unconstrained mirror-descent step,

$$\langle \boldsymbol{l}''_{t,h^\dagger} + \nabla F_{\xi(h^\dagger)}(\boldsymbol{p}_{t+1,h^\dagger}) - \nabla F_{\xi(h^\dagger)}(\boldsymbol{p}_{t,h^\dagger}), \, \boldsymbol{p} - \boldsymbol{p}_{t+1,h^\dagger} \rangle \geq 0.$$

This gives

$$\langle \boldsymbol{p}_{t+1,h^\dagger} - \boldsymbol{p}, \boldsymbol{l}''_{t,h^\dagger} \rangle \leq \langle \nabla F_{\xi(h^\dagger)}(\boldsymbol{p}_{t+1,h^\dagger}) - \nabla F_{\xi(h^\dagger)}(\boldsymbol{p}_{t,h^\dagger}), \, \boldsymbol{p} - \boldsymbol{p}_{t+1,h^\dagger} \rangle.$$

Using the definition of Bregman divergence,

$$\langle \nabla F_{\xi(h^\dagger)}(\boldsymbol{p}_{t+1,h^\dagger}) - \nabla F_{\xi(h^\dagger)}(\boldsymbol{p}_{t,h^\dagger}), \, \boldsymbol{p} - \boldsymbol{p}_{t+1,h^\dagger} \rangle = D_{F_{\xi(h^\dagger)}}(\boldsymbol{p}, \boldsymbol{p}_{t,h^\dagger}) - D_{F_{\xi(h^\dagger)}}(\boldsymbol{p}, \boldsymbol{p}_{t+1,h^\dagger}) - D_{F_{\xi(h^\dagger)}}(\boldsymbol{p}_{t+1,h^\dagger}, \boldsymbol{p}_{t,h^\dagger}),$$

we obtain

$$\langle \boldsymbol{p}_{t+1,h^\dagger} - \boldsymbol{p}, \boldsymbol{l}''_{t,h^\dagger} \rangle \leq D_{F_{\xi(h^\dagger)}}(\boldsymbol{p}, \boldsymbol{p}_{t,h^\dagger}) - D_{F_{\xi(h^\dagger)}}(\boldsymbol{p}, \boldsymbol{p}_{t+1,h^\dagger}) - D_{F_{\xi(h^\dagger)}}(\boldsymbol{p}_{t+1,h^\dagger}, \boldsymbol{p}_{t,h^\dagger}). \tag{18}$$

We now decompose

$$\langle \boldsymbol{p}_{t,h^\dagger} - \boldsymbol{p}, \boldsymbol{l}''_{t,h^\dagger} \rangle = \langle \boldsymbol{p}_{t,h^\dagger} - \boldsymbol{p}_{t+1,h^\dagger}, \boldsymbol{l}''_{t,h^\dagger} \rangle + \langle \boldsymbol{p}_{t+1,h^\dagger} - \boldsymbol{p}, \boldsymbol{l}''_{t,h^\dagger} \rangle.$$

Combining with equation 18 yields

$$\langle \boldsymbol{p}_{t,h^\dagger} - \boldsymbol{p}, \boldsymbol{l}''_{t,h^\dagger} \rangle \leq \langle \boldsymbol{p}_{t,h^\dagger} - \boldsymbol{p}_{t+1,h^\dagger}, \boldsymbol{l}''_{t,h^\dagger} \rangle - D_{F_{\xi(h^\dagger)}}(\boldsymbol{p}_{t+1,h^\dagger}, \boldsymbol{p}_{t,h^\dagger}) + D_{F_{\xi(h^\dagger)}}(\boldsymbol{p}, \boldsymbol{p}_{t,h^\dagger}) - D_{F_{\xi(h^\dagger)}}(\boldsymbol{p}, \boldsymbol{p}_{t+1,h^\dagger}). \tag{19}$$

Recall the unconstrained mirror step

$$\tilde{\boldsymbol{p}}_{t+1,h^\dagger} = \arg\min_{\boldsymbol{u}} \left\{ \langle \boldsymbol{l}''_{t,h^\dagger}, \boldsymbol{u} \rangle + D_{F_{\xi(h^\dagger)}}(\boldsymbol{u}, \boldsymbol{p}_{t,h^\dagger}) \right\}.$$

By the first-order optimality condition,

$$\boldsymbol{l}''_{t,h^\dagger} + \nabla F_{\xi(h^\dagger)}(\tilde{\boldsymbol{p}}_{t+1,h^\dagger}) - \nabla F_{\xi(h^\dagger)}(\boldsymbol{p}_{t,h^\dagger}) = \boldsymbol{0}. \tag{20}$$

Taking the inner product of equation 20 with $\boldsymbol{p}_{t,h^\dagger} - \boldsymbol{p}_{t+1,h^\dagger}$ yields

$$\langle \boldsymbol{p}_{t,h^\dagger} - \boldsymbol{p}_{t+1,h^\dagger}, \boldsymbol{l}''_{t,h^\dagger} \rangle = \langle \boldsymbol{p}_{t,h^\dagger} - \boldsymbol{p}_{t+1,h^\dagger}, \nabla F_{\xi(h^\dagger)}(\boldsymbol{p}_{t,h^\dagger}) - \nabla F_{\xi(h^\dagger)}(\tilde{\boldsymbol{p}}_{t+1,h^\dagger}) \rangle. \tag{21}$$

From the above, by using the definition of Bregman divergences, we have

$$\begin{aligned}
\langle \boldsymbol{p}_{t,h^\dagger} - \boldsymbol{p}_{t+1,h^\dagger}, \boldsymbol{l}''_{t,h^\dagger} \rangle &= \langle \boldsymbol{p}_{t,h^\dagger} - \boldsymbol{p}_{t+1,h^\dagger}, \nabla F_{\xi(h^\dagger)}(\boldsymbol{p}_{t,h^\dagger}) - \nabla F_{\xi(h^\dagger)}(\tilde{\boldsymbol{p}}_{t+1,h^\dagger}) \rangle \\
&= D_{F_{\xi(h^\dagger)}}(\boldsymbol{p}_{t+1,h^\dagger}, \boldsymbol{p}_{t,h^\dagger}) + D_{F_{\xi(h^\dagger)}}(\boldsymbol{p}_{t,h^\dagger}, \tilde{\boldsymbol{p}}_{t+1,h^\dagger}) - D_{F_{\xi(h^\dagger)}}(\boldsymbol{p}_{t+1,h^\dagger}, \tilde{\boldsymbol{p}}_{t+1,h^\dagger}) \\
&\leq D_{F_{\xi(h^\dagger)}}(\boldsymbol{p}_{t+1,h^\dagger}, \boldsymbol{p}_{t,h^\dagger}) + D_{F_{\xi(h^\dagger)}}(\boldsymbol{p}_{t,h^\dagger}, \tilde{\boldsymbol{p}}_{t+1,h^\dagger}). \tag{22}
\end{aligned}$$

Then from equation 19 and equation 22, we have

$$\langle \boldsymbol{p}_{t,h^\dagger} - \boldsymbol{p}, \boldsymbol{l}''_{t,h^\dagger} \rangle \leq D_{F_{\xi(h^\dagger)}}(\boldsymbol{p}, \boldsymbol{p}_{t,h^\dagger}) - D_{F_{\xi(h^\dagger)}}(\boldsymbol{p}, \boldsymbol{p}_{t+1,h^\dagger}) + D_{F_{\xi(h^\dagger)}}(\boldsymbol{p}_{t,h^\dagger}, \tilde{\boldsymbol{p}}_{t+1,h^\dagger}).$$

Summing over $t = t_s, \ldots, t_{s+1} - 1$ gives a telescoping series in the middle terms:

$$\sum_{t=t_s}^{t_{s+1}-1} \langle \boldsymbol{p}_{t,h^\dagger} - \boldsymbol{p}, \boldsymbol{l}''_{t,h^\dagger} \rangle \leq D_{F_{\xi(h^\dagger)}}(\boldsymbol{p}, \boldsymbol{p}_{t_s,h^\dagger}) - D_{F_{\xi(h^\dagger)}}(\boldsymbol{p}, \boldsymbol{p}_{t_{s+1},h^\dagger}) + \sum_{t=t_s}^{t_{s+1}} D_{F_{\xi(h^\dagger)}}(\boldsymbol{p}_{t,h^\dagger}, \tilde{\boldsymbol{p}}_{t+1,h^\dagger}).$$

Since $D_{F_{\xi(h^\dagger)}}(\boldsymbol{p}, \boldsymbol{p}_{t_{s+1},h^\dagger}) \geq 0$ by the nonnegativity of Bregman divergences, the term can be safely dropped. Therefore,

$$\sum_{t=t_s}^{t_{s+1}-1} \langle \boldsymbol{p}_{t,h^\dagger} - \boldsymbol{p}, \boldsymbol{l}''_{t,h^\dagger} \rangle \leq D_{F_{\xi(h^\dagger)}}(\boldsymbol{p}, \boldsymbol{p}_{t_s,h^\dagger}) + \sum_{t=t_s}^{t_{s+1}-1} D_{F_{\xi(h^\dagger)}}(\boldsymbol{p}_{t,h^\dagger}, \tilde{\boldsymbol{p}}_{t+1,h^\dagger}),$$

which concludes the proof. $\qquad\square$

In Lemma A.1, the first term is for the initial point diameter at time $t_s$ and the second term is for the divergence of the updated policy. Using the definition of the Bregman divergence and the fact that $p_{t_s,h^\dagger}(k) \geq \beta$, the initial point diameter term can be shown to be bounded as follows:

$$\begin{aligned}
D_{F_{\xi(h^\dagger)}}(\boldsymbol{p}, \boldsymbol{p}_{t_s,h^\dagger}) &\leq \frac{1}{\xi(h^\dagger)} \sum_{k \in [K]} p(k) \log(1/p_{t_s,h^\dagger}(k)) \\
&\leq \frac{\log(1/\beta)}{\xi(h^\dagger)}.
\end{aligned} \tag{23}$$

Next, for the updated policy divergence term, using $\tilde{p}_{t+1,h^\dagger}(k) = p_{t,h^\dagger}(k) \exp(-\xi(h^\dagger) l''_{t,h^\dagger}(k))$ for all $k \in [K]$, we have

$$\begin{aligned}
\sum_{t=t_s}^{t_{s+1}-1} & \mathbb{E}\left[D_{F_{\xi(h^\dagger)}}(\boldsymbol{p}_{t,h^\dagger}, \tilde{\boldsymbol{p}}_{t+1,h^\dagger})\right] \\
&= \sum_{t=t_s}^{t_{s+1}-1} \sum_{k=1}^{K} \mathbb{E}\left[\frac{1}{\xi(h^\dagger)} p_{t,h^\dagger}(k) \left(\exp(-\xi(h^\dagger) l''_{t,h^\dagger}(k)) - 1 + \xi(h^\dagger) l''_{t,h^\dagger}(k)\right)\right] \\
&\leq \sum_{t=t_s}^{t_{s+1}-1} \sum_{k=1}^{K} \mathbb{E}\left[\frac{\xi(h^\dagger)}{2} p_{t,h^\dagger}(k) l''_{t,h^\dagger}(k)^2\right] \\
&\leq \sum_{t=t_s}^{t_{s+1}-1} \sum_{k=1}^{K} \mathbb{E}\left[\frac{\xi(h^\dagger)}{2 p_t(h^\dagger)}\right] \leq \frac{\xi(h^\dagger) K T_s}{2\alpha},
\end{aligned} \tag{24}$$

where the first inequality comes from $\exp(-x) \leq 1 - x + x^2/2$ for all $x \geq 0$, the second inequality comes from $\mathbb{E}[l''_{t,h^\dagger}(k)^2 \mid p_{t,h^\dagger}(k), p_t(h^\dagger)] \leq 1/(p_t(h^\dagger) p_{t,h^\dagger}(k))$, and the last inequality is obtained from $p_t(h^\dagger) \geq \alpha$ from the clipped domain. We can observe that the clipped domain controls the variance of estimators. Then from equation 17, Lemma A.1, equation 23, and equation 24, by summing up over $s \in [S]$, we have

$$\sum_{t=1}^{T} \mathbb{E}\left[l_t(a_{t,h^\dagger})\right] - \sum_{s=0}^{S} \min_{k_s \in [K]} \sum_{t=t_s}^{t_{s+1}-1} l_t(k_s) \leq \beta T(K-1) + \frac{S \log(1/\beta)}{\xi(h^\dagger)} + \frac{\xi(h^\dagger) K T}{2\alpha}. \tag{25}$$

Next, we provide a bound for the following regret from the master:

$$\sum_{t=1}^{T} \mathbb{E}\left[l_t(a_{t,h_t})\right] - \sum_{t=1}^{T} \mathbb{E}\left[l_t(a_{t,h^\dagger})\right].$$

Let $\tilde{\boldsymbol{p}}_{t+1} = \arg\min_{\boldsymbol{p}\in\mathbb{R}^H} \langle \boldsymbol{p}, \boldsymbol{l}_t'\rangle + D_{F_\eta}(\boldsymbol{p}, \boldsymbol{p}_t)$ and $\boldsymbol{e}_{h,H}$ denote the unit vector with 1 at base $h$-index and 0 at the rest of $H-1$ indices. For ease of presentation, we define $\tilde{l}_t(h) = l_t(a_{t,h})$ for $h \in [H]$. Then, we have

$$\sum_{t=1}^T \mathbb{E}\left[l_t(a_{t,h_t}) - l_t(a_{t,h^\dagger})\right] = \sum_{t=1}^T \mathbb{E}\left[\langle \boldsymbol{p}_t - \boldsymbol{e}_{h^\dagger,H}, \tilde{\boldsymbol{l}}_t\rangle\right]$$

$$\leq \max_{\boldsymbol{p}\in\mathcal{P}_H^\alpha} \mathbb{E}\left[\sum_{t=1}^T \langle \boldsymbol{p} - \boldsymbol{e}_{h^\dagger,H}, \tilde{\boldsymbol{l}}_t\rangle + \sum_{t=1}^T \langle \boldsymbol{p}_t - \boldsymbol{p}, \tilde{\boldsymbol{l}}_t\rangle\right]$$

$$\leq \alpha T(H-1) + \max_{\boldsymbol{p}\in\mathcal{P}_H^\alpha} \mathbb{E}\left[\sum_{t=1}^T \langle \boldsymbol{p}_t - \boldsymbol{p}, \tilde{\boldsymbol{l}}_t\rangle\right]. \tag{26}$$

For bounding the second term in equation 26, we use the following lemma.

**Lemma A.2** (Theorem 28.4 and Eq. 28.11 in Lattimore & Szepesvári (2020)). *For any $\boldsymbol{p} \in \mathcal{P}_H^\alpha$ we have*

$$\mathbb{E}\left[\sum_{t=1}^T \langle \boldsymbol{p}_t - \boldsymbol{p}, \tilde{\boldsymbol{l}}_t\rangle\right] \leq \mathbb{E}\left[D_{F_\eta}(\boldsymbol{p}, \boldsymbol{p}_1) + \sum_{t=1}^T D_{F_\eta}(\boldsymbol{p}_t, \tilde{\boldsymbol{p}}_{t+1})\right].$$

*Proof.* For completeness, we provide a proof for this lemma. Fix any $\boldsymbol{p} \in \mathcal{P}_H^\alpha$. By the first-order optimality condition of the unconstrained mirror-descent step,

$$\langle \boldsymbol{l}_t' + \nabla F_\eta(\boldsymbol{p}_{t+1}) - \nabla F_\eta(\boldsymbol{p}_t),\, \boldsymbol{p} - \boldsymbol{p}_{t+1}\rangle \geq 0.$$

This gives

$$\langle \boldsymbol{p}_{t+1} - \boldsymbol{p}, \boldsymbol{l}_t'\rangle \leq \langle \nabla F_\eta(\boldsymbol{p}_{t+1}) - \nabla F_\eta(\boldsymbol{p}_t),\, \boldsymbol{p} - \boldsymbol{p}_{t+1}\rangle.$$

Using the three-point identity for Bregman divergences,

$$\langle \nabla F_\eta(\boldsymbol{p}_{t+1}) - \nabla F_\eta(\boldsymbol{p}_t),\, \boldsymbol{p} - \boldsymbol{p}_{t+1}\rangle = D_{F_\eta}(\boldsymbol{p}, \boldsymbol{p}_t) - D_{F_\eta}(\boldsymbol{p}, \boldsymbol{p}_{t+1}) - D_{F_\eta}(\boldsymbol{p}_{t+1}, \boldsymbol{p}_t),$$

we obtain

$$\langle \boldsymbol{p}_{t+1} - \boldsymbol{p}, \boldsymbol{l}_t'\rangle \leq D_{F_\eta}(\boldsymbol{p}, \boldsymbol{p}_t) - D_{F_\eta}(\boldsymbol{p}, \boldsymbol{p}_{t+1}) - D_{F_\eta}(\boldsymbol{p}_{t+1}, \boldsymbol{p}_t). \tag{27}$$

We now decompose

$$\langle \boldsymbol{p}_t - \boldsymbol{p}, \boldsymbol{l}_t'\rangle = \langle \boldsymbol{p}_t - \boldsymbol{p}_{t+1}, \boldsymbol{l}_t'\rangle + \langle \boldsymbol{p}_{t+1} - \boldsymbol{p}, \boldsymbol{l}_t'\rangle.$$

Combining with equation 27 yields

$$\langle \boldsymbol{p}_t - \boldsymbol{p}, \boldsymbol{l}_t'\rangle \leq \langle \boldsymbol{p}_t - \boldsymbol{p}_{t+1}, \boldsymbol{l}_t'\rangle - D_{F_\eta}(\boldsymbol{p}_{t+1}, \boldsymbol{p}_t) + D_{F_\eta}(\boldsymbol{p}, \boldsymbol{p}_t) - D_{F_\eta}(\boldsymbol{p}, \boldsymbol{p}_{t+1}). \tag{28}$$

Recall the unconstrained mirror step

$$\tilde{\boldsymbol{p}}_{t+1} = \arg\min_{\boldsymbol{u}}\left\{\langle \boldsymbol{l}_t', \boldsymbol{u}\rangle + D_{F_\eta}(\boldsymbol{u}, \boldsymbol{p}_t)\right\}.$$

By the first-order optimality condition,

$$\boldsymbol{l}_t' + \nabla F_\eta(\tilde{\boldsymbol{p}}_{t+1}) - \nabla F_\eta(\boldsymbol{p}_t) = \boldsymbol{0}. \tag{29}$$

Taking the inner product of equation 29 with $\boldsymbol{p}_t - \boldsymbol{p}_{t+1}$ yields

$$\langle \boldsymbol{p}_t - \boldsymbol{p}_{t+1},\, \boldsymbol{l}_t'\rangle = \langle \boldsymbol{p}_t - \boldsymbol{p}_{t+1},\, \nabla F_\eta(\boldsymbol{p}_t) - \nabla F_\eta(\tilde{\boldsymbol{p}}_{t+1})\rangle. \tag{30}$$

From the above, by using the definition of Bregman divergences, we have

$$\begin{aligned}
\langle \boldsymbol{p}_t - \boldsymbol{p}_{t+1},\, \boldsymbol{l}_t'\rangle &= \langle \boldsymbol{p}_t - \boldsymbol{p}_{t+1},\, \nabla F_\eta(\boldsymbol{p}_t) - \nabla F_\eta(\tilde{\boldsymbol{p}}_{t+1})\rangle \\
&= D_{F_\eta}(\boldsymbol{p}_{t+1}, \boldsymbol{p}_t) + D_{F_\eta}(\boldsymbol{p}_t, \tilde{\boldsymbol{p}}_{t+1}) - D_{F_\eta}(\boldsymbol{p}_{t+1}, \tilde{\boldsymbol{p}}_{t+1}) \\
&\leq D_{F_\eta}(\boldsymbol{p}_{t+1}, \boldsymbol{p}_t) + D_{F_\eta}(\boldsymbol{p}_t, \tilde{\boldsymbol{p}}_{t+1}).
\end{aligned} \tag{31}$$

Then from equation 28 and equation 31, we have

$$\langle \boldsymbol{p}_t - \boldsymbol{p}, \boldsymbol{l}_t' \rangle \leq D_{F_\eta}(\boldsymbol{p}, \boldsymbol{p}_t) - D_{F_\eta}(\boldsymbol{p}, \boldsymbol{p}_{t+1}) + D_{F_\eta}(\boldsymbol{p}_t, \tilde{\boldsymbol{p}}_{t+1}).$$

Summing over $t = 1, \ldots, T$ gives a telescoping series in the middle terms:

$$\sum_{t=1}^{T} \langle \boldsymbol{p}_t - \boldsymbol{p}, \boldsymbol{l}_t' \rangle \leq D_{F_\eta}(\boldsymbol{p}, \boldsymbol{p}_1) - D_{F_\eta}(\boldsymbol{p}, \boldsymbol{p}_{T+1}) + \sum_{t=1}^{T} D_{F_\eta}(\boldsymbol{p}_t, \tilde{\boldsymbol{p}}_{t+1}).$$

Since $D_{F_\eta}(\boldsymbol{p}, \boldsymbol{p}_{T+1}) \geq 0$ by the nonnegativity of Bregman divergences, the term can be safely dropped. Therefore,

$$\sum_{t=1}^{T} \langle \boldsymbol{p}_t - \boldsymbol{p}, \boldsymbol{l}_t' \rangle \leq D_{F_\eta}(\boldsymbol{p}, \boldsymbol{p}_1) + \sum_{t=1}^{T} D_{F_\eta}(\boldsymbol{p}_t, \tilde{\boldsymbol{p}}_{t+1}),$$

which concludes the proof with $\mathbb{E}[\sum_{t=1}^{T} \langle \boldsymbol{p}_t - \boldsymbol{p}, \boldsymbol{l}_t' \rangle] = \mathbb{E}[\sum_{t=1}^{T} \langle \boldsymbol{p}_t - \boldsymbol{p}, \tilde{\boldsymbol{l}}_t \rangle].$ □

From equation 26 and Lemma A.2, we have

$$\sum_{t=1}^{T} \mathbb{E}\left[l_t(a_{t,h_t})\right] - \sum_{t=1}^{T} \mathbb{E}\left[l_t(a_{t,h^\dagger})\right]$$

$$\leq \alpha T(H-1) + \max_{\boldsymbol{p} \in \mathcal{P}_H^\alpha} \mathbb{E}\left[D_{F_\eta}(\boldsymbol{p}, \boldsymbol{p}_1) + \sum_{t=1}^{T} D_{F_\eta}(\boldsymbol{p}_t, \tilde{\boldsymbol{p}}_{t+1})\right]$$

$$\leq \alpha T(H-1) + \frac{\log(H)}{\eta} + \frac{\eta T H}{2}, \tag{32}$$

where the last inequality is obtained from the fact that

$$D_{F_\eta}(\boldsymbol{p}, \boldsymbol{p}_1) = \frac{1}{\eta} \sum_{h \in [H]} p(h) \log\left(\frac{p(h)}{p_1(h)}\right) \leq \frac{\log(H)}{\eta}$$

and from

$$\mathbb{E}\left[\sum_{t=1}^{T} D_{F_\eta}(\boldsymbol{p}_t, \tilde{\boldsymbol{p}}_{t+1})\right] = \mathbb{E}\left[(1/\eta) \sum_{t=1}^{T} \sum_{h \in [H]} p_t(h)(\exp(-\eta l_t'(h)) - 1 + \eta l_t'(h))\right]$$

$$\leq \frac{\eta}{2} \mathbb{E}\left[\sum_{t=1}^{T} \sum_{h \in [H]} p_t(h) l_t'(h)^2\right] \leq \frac{\eta T H}{2}.$$

Therefore, putting equation 16, equation 25, and equation 32 altogether, we have

$$R_S(T) = \sum_{t=1}^{T} \mathbb{E}\left[l_t(a_t)\right] - \sum_{s=0}^{S} \min_{1 \leq k_s \leq K} \sum_{t=T_s}^{T_{s+1}-1} l_t(k_s)$$

$$\leq \alpha T H + \frac{\log(H)}{\eta} + \frac{\eta T H}{2} + \beta T(K-1) + \frac{S \log(1/\beta)}{\xi(h^\dagger)} + \frac{\xi(h^\dagger) K T}{2\alpha}$$

$$= \tilde{O}(S^{1/2} T^{2/3} K^{1/3}),$$

where $\alpha = K^{1/3}/(T^{1/3} H^{1/2})$, $\beta = 1/(KT)$, $\eta = 1/\sqrt{TH}$, $\xi(h^\dagger) = (h^\dagger)^{1/2}/(K^{1/3} T^{2/3})$, $h^\dagger = \Theta(S)$, and $H = \log(T)$. This concludes the proof.

## A.2 Proof of Theorem 3.2

Let $t_s$ be the time when the $s$-th switch of the best arm happens and $t_{S+1} - 1 = T$, $t_0 = 1$. Also let $t_{s+1} - t_s = T_s$. For any $t_s$ for all $s \in [0, S]$, the $S$-switch regret can be expressed as

$$R_S(T) = \sum_{t=1}^{T} \mathbb{E}\left[l_t(a_t)\right] - \sum_{s=0}^{S} \min_{k_s \in [K]} \sum_{t=t_s}^{t_{s+1}-1} l_t(k_s)$$

$$= \sum_{t=1}^{T} \mathbb{E}\left[l_t(a_{t,h_t})\right] - \sum_{t=1}^{T} \mathbb{E}\left[l_t(a_{t,h^\dagger})\right] + \sum_{t=1}^{T} \mathbb{E}\left[l_t(a_{t,h^\dagger})\right] - \sum_{s=0}^{S} \min_{k_s \in [K]} \sum_{t=t_s}^{t_{s+1}-1} l_t(k_s), \tag{33}$$

in which the first two terms are closely related with the regret from the master algorithm against the near optimal base $h^\dagger$, and the remaining terms are related with the regret from $h^\dagger$ base algorithm against the best arms in hindsight.

First we provide a bound for the following regret from base $h^\dagger$. From equation 17, we can obtain

$$\sum_{t=t_s}^{t_{s+1}-1} \mathbb{E}\left[l_t(a_{t,h^\dagger})\right] - \min_{k_s \in [K]} \sum_{t=t_s}^{t_{s+1}-1} l_t(k_s) \le \beta T_s K + \max_{p \in \mathcal{P}_K^\beta} \mathbb{E}\left[\sum_{t=t_s}^{t_{s+1}-1} \langle \boldsymbol{p}_{t,h^\dagger} - \boldsymbol{p}, \boldsymbol{l}''_{t,h^\dagger}\rangle\right]. \tag{34}$$

Then for the second term of the last inequality in equation 34, we provide a following lemma.

**Lemma A.3** (Restatement of Lemma 3.3). *For any $\boldsymbol{p} \in \mathcal{P}_K^\beta$ we can show that*

$$\sum_{t=t_s}^{t_{s+1}-1} \mathbb{E}\left[\langle \boldsymbol{p}_{t,h^\dagger} - \boldsymbol{p}, \boldsymbol{l}''_{t,h^\dagger}\rangle\right] \le \mathbb{E}\left[2\log(1/\beta)\sqrt{\frac{KT\rho_T(h^\dagger)}{h^\dagger}} + \frac{T_s}{2}\sqrt{\frac{SK\rho_T(h^\dagger)}{T}}\right].$$

*Proof.* For ease of presentation, we define the negative entropy regularizer without a learning rate as

$$F(\boldsymbol{p}) = \sum_{i=1}^{K} (p(i)\log p(i) - p(i))$$

and define learning rate $\xi_0(h^\dagger) = \infty$. From the first-order optimality condition for $\boldsymbol{p}_{t+1,h^\dagger}$ and using the definition of the Bregman divergence,

$$\langle \boldsymbol{p}_{t+1,h^\dagger} - \boldsymbol{p}, \boldsymbol{l}''_{t,h^\dagger}\rangle$$

$$\le \frac{1}{\xi_t(h^\dagger)}\langle \boldsymbol{p} - \boldsymbol{p}_{t+1,h^\dagger}, \nabla F(\boldsymbol{p}_{t+1,h^\dagger}) - \nabla F(\boldsymbol{p}_{t,h^\dagger})\rangle$$

$$= \frac{1}{\xi_t(h^\dagger)}\left(D_F(\boldsymbol{p}, \boldsymbol{p}_{t,h^\dagger}) - D_F(\boldsymbol{p}, \boldsymbol{p}_{t+1,h^\dagger}) - D_F(\boldsymbol{p}_{t+1,h^\dagger}, \boldsymbol{p}_{t,h^\dagger})\right). \tag{35}$$

Also, we have

$$\langle \boldsymbol{p}_{t,h^\dagger} - \boldsymbol{p}_{t+1,h^\dagger}, \boldsymbol{l}''_{t,h^\dagger}\rangle$$

$$= \frac{1}{\xi_t(h^\dagger)}\langle \boldsymbol{p}_{t,h^\dagger} - \boldsymbol{p}_{t+1,h^\dagger}, \nabla F(\boldsymbol{p}_{t,h^\dagger}) - \nabla F(\tilde{\boldsymbol{p}}_{t+1,h^\dagger})\rangle$$

$$= \frac{1}{\xi_t(h^\dagger)}(D_F(\boldsymbol{p}_{t+1,h^\dagger}, \boldsymbol{p}_{t,h^\dagger}) + D_F(\boldsymbol{p}_{t,h^\dagger}, \tilde{\boldsymbol{p}}_{t+1,h^\dagger}) - D_F(\boldsymbol{p}_{t+1,h^\dagger}, \tilde{\boldsymbol{p}}_{t+1,h^\dagger}))$$

$$\le \frac{1}{\xi_t(h^\dagger)}(D_F(\boldsymbol{p}_{t+1,h^\dagger}, \boldsymbol{p}_{t,h^\dagger}) + D_F(\boldsymbol{p}_{t,h^\dagger}, \tilde{\boldsymbol{p}}_{t+1,h^\dagger})). \tag{36}$$

Then, we can obtain

$$
\begin{aligned}
\sum_{t=t_s}^{t_{s+1}-1} \langle \boldsymbol{p}_{t,h^\dagger} - \boldsymbol{p}, \boldsymbol{l}''_{t,h^\dagger} \rangle \leq{} & \sum_{t=t_s}^{t_{s+1}-1} \langle \boldsymbol{p}_{t,h^\dagger} - \boldsymbol{p}_{t+1,h^\dagger}, \boldsymbol{l}''_{t,h^\dagger} \rangle \\
& + \sum_{t=t_s}^{t_{s+1}-1} \frac{1}{\xi_t(h^\dagger)} \left( D_F(\boldsymbol{p}, \boldsymbol{p}_{t,h^\dagger}) - D_F(\boldsymbol{p}, \boldsymbol{p}_{t+1,h^\dagger}) - D_F(\boldsymbol{p}_{t+1,h^\dagger}, \boldsymbol{p}_{t,h^\dagger}) \right) \\
={} & \sum_{t=t_s}^{t_{s+1}-1} \langle \boldsymbol{p}_{t,h^\dagger} - \boldsymbol{p}_{t+1,h^\dagger}, \boldsymbol{l}''_{t,h^\dagger} \rangle + \sum_{t=t_s+1}^{t_{s+1}-1} D_F(\boldsymbol{p}, \boldsymbol{p}_{t,h^\dagger}) \left( \frac{1}{\xi_t(h^\dagger)} - \frac{1}{\xi_{t-1}(h^\dagger)} \right) \\
& + \frac{1}{\xi_{t_s}(h^\dagger)} D_F(\boldsymbol{p}, \boldsymbol{p}_{t_s,h^\dagger}) - \frac{1}{\xi_{t_{s+1}-1}(h)} D_F(\boldsymbol{p}, \boldsymbol{p}_{t_{s+1},h^\dagger}) - \sum_{t=t_s}^{t_{s+1}-1} \frac{1}{\xi_t(h^\dagger)} D_F(\boldsymbol{p}_{t+1,h^\dagger}, \boldsymbol{p}_{t,h^\dagger}) \\
\leq{} & \sum_{t=t_s}^{t_{s+1}-1} \langle \boldsymbol{p}_{t,h^\dagger} - \boldsymbol{p}_{t+1,h^\dagger}, \boldsymbol{l}''_{t,h^\dagger} \rangle + \log(1/\beta) \sum_{t=t_s+1}^{t_{s+1}-1} \left( \frac{1}{\xi_t(h^\dagger)} - \frac{1}{\xi_{t-1}(h^\dagger)} \right) \\
& + \frac{1}{\xi_{t_s}(h^\dagger)} D_F(\boldsymbol{p}, \boldsymbol{p}_{t_s,h^\dagger}) - \frac{1}{\xi_{t_{s+1}-1}(h)} D_F(\boldsymbol{p}, \boldsymbol{p}_{t_{s+1},h^\dagger}) - \sum_{t=t_s}^{t_{s+1}-1} \frac{1}{\xi_t(h^\dagger)} D_F(\boldsymbol{p}_{t+1,h^\dagger}, \boldsymbol{p}_{t,h^\dagger}) \\
\leq{} & 2 \frac{\log(1/\beta)}{\xi_T(h^\dagger)} + \sum_{t=t_s}^{t_{s+1}-1} \frac{D_F(\boldsymbol{p}_{t,h^\dagger}, \tilde{\boldsymbol{p}}_{t+1,h^\dagger})}{\xi_t(h^\dagger)} \\
={} & 2 \log(1/\beta) \sqrt{\frac{KT\rho_T(h^\dagger)}{h^\dagger}} + \sum_{t=t_s}^{t_{s+1}-1} \frac{D_F(\boldsymbol{p}_{t,h^\dagger}, \tilde{\boldsymbol{p}}_{t+1,h^\dagger})}{\xi_t(h^\dagger)},
\end{aligned}
\tag{37}
$$

where the first inequality is obtained from equation 35, the second last inequality is obtained from $D_F(\boldsymbol{p}, \boldsymbol{p}_{t,h^\dagger}) \leq \log(1/\beta)$ and $1/\xi_t(h^\dagger) \geq 1/\xi_{t-1}(h^\dagger)$ from non-decreasing $\rho_t(h^\dagger)$, and the last inequality is obtained from equation 36, $D_F(\boldsymbol{p}, \boldsymbol{p}_{t_s,h^\dagger}) \leq \log(1/\beta)$, and $\xi_s(h^\dagger) \geq \xi_T(h^\dagger)$ for $s \in [T]$ from non-decreasing $\rho_s(h^\dagger)$.

For the second term in the last inequality in equation 37, using $\tilde{p}_{t+1,h^\dagger}(k) = p_{t,h^\dagger}(k) \exp(-\xi(h^\dagger) l''_{t,h^\dagger}(k))$ for all $k \in [K]$, we have

$$
\begin{aligned}
\sum_{t=t_s}^{t_{s+1}-1} \mathbb{E} \left[ \frac{D_F(\boldsymbol{p}_{t,h^\dagger}, \tilde{\boldsymbol{p}}_{t+1,h^\dagger})}{\xi_t(h^\dagger)} \right] ={} & \sum_{t=t_s}^{t_{s+1}-1} \sum_{k=1}^{K} \mathbb{E} \left[ \frac{1}{\xi_t(h^\dagger)} p_{t,h^\dagger}(k) \Big( \exp(-\xi_t(h^\dagger) l''_{t,h^\dagger}(k)) \right. \\
& \left. \hspace{6cm} -1 + \xi_t(h^\dagger) l''_{t,h^\dagger}(k) \Big) \right] \\
\leq{} & \sum_{t=t_s}^{t_{s+1}-1} \sum_{k=1}^{K} \mathbb{E} \left[ \frac{\xi_t(h^\dagger)}{2} p_{t,h^\dagger}(k) l''_{t,h^\dagger}(k)^2 \right] \\
\leq{} & \sum_{t=t_s}^{t_{s+1}-1} \sum_{k=1}^{K} \mathbb{E} \left[ \frac{\xi_t(h^\dagger)}{2 p_t(h^\dagger)} \right] \\
\leq{} & \sum_{t=t_s}^{t_{s+1}-1} \sum_{k=1}^{K} \mathbb{E} \left[ \frac{\xi_t(h^\dagger) \rho_t(h^\dagger)}{2} \right] \\
\leq{} & \sum_{t=t_s}^{t_{s+1}-1} \sum_{k=1}^{K} \mathbb{E} \left[ \frac{1}{2} \sqrt{\frac{h^\dagger \rho_t(h^\dagger)}{KT}} \right] \\
\leq{} & T_s \sqrt{\frac{h^\dagger K}{T}} \frac{\mathbb{E} \left[ \rho_T(h^\dagger)^{1/2} \right]}{2},
\end{aligned}
\tag{38}
$$

where the first inequality comes from $\exp(-x) \leq 1 - x + x^2/2$ for all $x \geq 0$, the second inequality comes from $\mathbb{E}[l''_{t,h^\dagger}(k)^2 \mid p_{t,h^\dagger}(k), p_t(h^\dagger)] \leq 1/(p_t(h^\dagger)p_{t,h^\dagger}(k))$, and the third inequality is obtained from $1/p_t(h^\dagger) \leq \rho_t(h^\dagger)$.

$\square$

Then from equation 34 and Lemma A.3, we have

$$
\sum_{t=1}^{T} \mathbb{E}\left[l_t(a_{t,h^\dagger})\right] - \sum_{s=0}^{S} \min_{1 \leq k_s \leq K} \sum_{t=T_s}^{T_{s+1}-1} l_t(k_s)
$$
$$
\leq \beta T(K-1) + \mathbb{E}\left[2S\log(1/\beta)\sqrt{\frac{KT\rho_T(h^\dagger)}{h^\dagger}} + \frac{1}{2}\sqrt{TSK\rho_T(h^\dagger)}\right]. \tag{39}
$$

Next, we provide a bound for the regret from the master in the following lemma.

**Lemma A.4** (Lemma 13 in Agarwal et al. (2017)).

$$
\sum_{t=1}^{T} \mathbb{E}\left[l_t(a_{t,h_t})\right] - \sum_{t=1}^{T} \mathbb{E}\left[l_t(a_{t,h^\dagger})\right] \leq O\left(\frac{H\log(T)}{\eta} + T\eta\right) - \mathbb{E}\left[\frac{\rho_T(h^\dagger)}{40\eta\log T}\right] + \alpha T(H-1).
$$

*Proof.* For ease of presentation, define $\tilde{l}_t(h) = l_t(a_{t,h})$ for $h \in [H]$. Then

$$
\sum_{t=1}^{T} \mathbb{E}\left[l_t(a_{t,h_t}) - l_t(a_{t,h^\dagger})\right] = \sum_{t=1}^{T} \mathbb{E}\left[\langle \boldsymbol{p}_t - \boldsymbol{e}_{h^\dagger,H}, \tilde{\boldsymbol{l}}_t\rangle\right]
$$
$$
\leq \max_{\boldsymbol{p} \in \mathcal{P}_H^\alpha} \mathbb{E}\left[\sum_{t=1}^{T}\langle \boldsymbol{p} - \boldsymbol{e}_{h^\dagger,H}, \tilde{\boldsymbol{l}}_t\rangle + \sum_{t=1}^{T}\langle \boldsymbol{p}_t - \boldsymbol{p}, \tilde{\boldsymbol{l}}_t\rangle\right]
$$
$$
\leq \alpha T(H-1) + \max_{\boldsymbol{p} \in \mathcal{P}_H^\alpha} \mathbb{E}\left[\sum_{t=1}^{T}\langle \boldsymbol{p}_t - \boldsymbol{p}, \tilde{\boldsymbol{l}}_t\rangle\right] \tag{40}
$$
$$
= \alpha T(H-1) + \max_{\boldsymbol{p} \in \mathcal{P}_H^\alpha} \mathbb{E}\left[\sum_{t=1}^{T}\langle \boldsymbol{p}_t - \boldsymbol{p}, \boldsymbol{l}'_t\rangle\right]. \tag{41}
$$

**Bounding the mirror-descent term.** We next bound $\mathbb{E}\left[\sum_t \langle \boldsymbol{p}_t - \boldsymbol{p}, \boldsymbol{l}'_t\rangle\right]$ using the OMD analysis of Agarwal et al. (2017, Lemma 13). The master update is

$$
\boldsymbol{p}_{t+1} = \arg\min_{\boldsymbol{p} \in \mathcal{P}_H^\alpha}\left\{\langle \boldsymbol{p}, \boldsymbol{l}'_t\rangle + D_{G_{\boldsymbol{\eta}_t}}(\boldsymbol{p}, \boldsymbol{p}_t)\right\},
$$

where $G_{\boldsymbol{\eta}_t}(\boldsymbol{p}) = \sum_{h=1}^{H} \frac{1}{\eta_t(h)} p(h)\log p(h)$ and $D_{G_{\boldsymbol{\eta}_t}}$ is the corresponding Bregman divergence. Applying the standard mirror-descent inequality, for any $\boldsymbol{p} \in \mathcal{P}_H^\alpha$,

$$
\langle \boldsymbol{p}_t - \boldsymbol{p}, \tilde{\boldsymbol{l}}_t\rangle \leq D_{G_{\boldsymbol{\eta}_t}}(\boldsymbol{p}, \boldsymbol{p}_t) - D_{G_{\boldsymbol{\eta}_t}}(\boldsymbol{p}, \boldsymbol{p}_{t+1}) + \sum_{h=1}^{H} \eta_t(h)\, p_t(h)^2\, l'_t(h)^2. \tag{42}
$$

Summing over $t = 1, \ldots, T$ gives

$$
\sum_{t=1}^{T}\langle \boldsymbol{p}_t - \boldsymbol{p}, \tilde{\boldsymbol{l}}_t\rangle \leq \sum_{t=1}^{T}\left(D_{G_{\boldsymbol{\eta}_t}}(\boldsymbol{p}, \boldsymbol{p}_t) - D_{G_{\boldsymbol{\eta}_t}}(\boldsymbol{p}, \boldsymbol{p}_{t+1})\right) + \sum_{t=1}^{T}\sum_{h=1}^{H} \eta_t(h)\, p_t(h)^2\, l'_t(h)^2. \tag{43}
$$

**Bounding the potential differences.** We first control the telescoping term in equation 43. Since

$D_{G_{\boldsymbol{\eta}_t}}(\cdot, \cdot) \geq 0$ and the learning rates $\eta_t(h)$ are nondecreasing in $t$ for each $h$, we have

$$\sum_{t=1}^{T}\Big(D_{G_{\boldsymbol{\eta}_t}}(\boldsymbol{p}, \boldsymbol{p}_t) - D_{G_{\boldsymbol{\eta}_t}}(\boldsymbol{p}, \boldsymbol{p}_{t+1})\Big) \leq D_{G_{\boldsymbol{\eta}_1}}(\boldsymbol{p}, \boldsymbol{p}_1) + \sum_{t=1}^{T-1}\sum_{h=1}^{H}\Big(\tfrac{1}{\eta_{t+1}(h)} - \tfrac{1}{\eta_t(h)}\Big) h\Big(\tfrac{p(h)}{p_{t+1}(h)}\Big), \qquad (44)$$

where $h(y) = y - 1 - \log y \geq 0$ is the log-barrier Bregman core. For the initial term, using that $G_{\boldsymbol{\eta}_1}$ is (scaled) negative entropy on the clipped simplex $\mathcal{P}_H^\alpha$ with $\alpha = 1/(TH)$, we obtain the standard bound

$$\max_{\boldsymbol{p} \in \mathcal{P}_H^\alpha} D_{G_{\boldsymbol{\eta}_1}}(\boldsymbol{p}, \boldsymbol{p}_1) = O\Big(\frac{H \log(1/\alpha)}{\eta}\Big) = O\Big(\frac{H \log T}{\eta}\Big). \qquad (45)$$

**Adaptive-rate gain (negative correction).** By the adaptive schedule in Algorithm 2, if $\frac{1}{p_{t+1}(h)} > \rho_t(h)$, then $\rho_{t+1}(h) = \frac{2}{p_{t+1}(h)}$ and $\eta_{t+1}(h) = \gamma\,\eta_t(h)$ with $\gamma = e^{1/\log T}$, while otherwise $\eta_{t+1}(h) = \eta_t(h)$. As in (Agarwal et al., 2017, Lemma 13), this implies that whenever the coordinate $h^\dagger$ is assigned too little probability, the factor $\big(\frac{1}{\eta_{t+1}(h^\dagger)} - \frac{1}{\eta_t(h^\dagger)}\big)$ is negative of order $-1/(\eta \log T)$, and it multiplies the nonnegative barrier increment $h\Big(\frac{p(h^\dagger)}{p_{t+1}(h^\dagger)}\Big)$ where $h(y) = y - 1 - \log(y)$. Aggregating these events over $t = 1, \ldots, T-1$ yields

$$\sum_{t=1}^{T-1}\sum_{h=1}^{H}\Big(\tfrac{1}{\eta_{t+1}(h)} - \tfrac{1}{\eta_t(h)}\Big) h\Big(\tfrac{p(h)}{p_{t+1}(h)}\Big) \leq \sum_{t=1}^{T-1}\Big(\tfrac{1}{\eta_{t+1}(h^\dagger)} - \tfrac{1}{\eta_t(h^\dagger)}\Big) h\Big(\tfrac{p(h^\dagger)}{p_{t+1}(h^\dagger)}\Big) \leq -\frac{\rho_T(h^\dagger)}{40\,\eta \log T}, \qquad (46)$$

where $\rho_T(h^\dagger)$ is the final density parameter maintained by the schedule. Combining equation 44, equation 45, and equation 46, and for $\boldsymbol{p} \in \mathcal{P}_H^\alpha$, we obtain

$$\sum_{t=1}^{T}\Big(D_{G_{\boldsymbol{\eta}_t}}(\boldsymbol{p}, \boldsymbol{p}_t) - D_{G_{\boldsymbol{\eta}_t}}(\boldsymbol{p}, \boldsymbol{p}_{t+1})\Big) \leq O\Big(\frac{H \log T}{\eta}\Big) - \frac{\rho_T(h^\dagger)}{40\,\eta \log T}. \qquad (47)$$

**Bounding the variance term.** It remains to bound $\sum_{t=1}^{T}\sum_{h=1}^{H}\eta_t(h)\,p_t(h)^2\,l_t'(h)^2$. Recall that $l_t'(h) \in [0, 1/p_t(h)]$ and only the sampled coordinate can be nonzero. Since each increase at least doubles the density $\rho_t(h)$ and $\rho_t(h) \leq 2TH$ from $p_t(h) \geq \alpha = 1/TH$, the number of entire updates for each $h$ is at most $C_1 \log(HT)$ for a constant $C_1 > 0$. This implies that $\eta_t(h) \leq \eta_T(h) \leq \eta\gamma^{C_1 \log(2HT)} \leq \eta e^{C_2}$ for a constant $C_2 > 0$. Therefore

$$\sum_{t=1}^{T}\sum_{h=1}^{H}\eta_t(h)\,p_t(h)^2\,l_t'(h)^2 = \sum_{t=1}^{T}\eta_t(h_t)\,p_t(h_t)^2\,l_t'(h_t)^2 \leq T\eta_T(h) = O(T\eta). \qquad (48)$$

**Putting the pieces together.** Apply equation 47 and equation 48 to equation 43, then maximize over $\boldsymbol{p} \in \mathcal{P}_H^\alpha$ and take expectations. Combining with equation 41 yields

$$\sum_{t=1}^{T}\mathbb{E}[l_t(a_{t,h_t})] - \sum_{t=1}^{T}\mathbb{E}\big[l_t(a_{t,h^\dagger})\big] \leq O\Big(\frac{H \log T}{\eta}\Big) + O(T\eta) - \mathbb{E}\Big[\frac{\rho_T(h^\dagger)}{40\,\eta \log T}\Big] + \alpha T(H - 1),$$

which is the desired bound. $\qquad\qquad\qquad\qquad\qquad\qquad\qquad\qquad\qquad\qquad\qquad\qquad\qquad\square$

The negative bias term in Lemma A.4 is derived from the log-barrier regularizer and increasing learning rates $\eta_t(h)$. This term is critical to bound the worst case regret which will be shown soon. Also, $H \log(T)/\eta$ is obtained from $H \log(1/(H\alpha))/\eta$ considering the clipped domain. Then, putting equation 33 and Lemmas A.3

and A.4 altogether, we have

$$
\begin{aligned}
R_S(T) &= \sum_{t=1}^{T} \mathbb{E}\left[l_t(a_t)\right] - \sum_{s=0}^{S} \min_{1 \le k_s \le K} \sum_{t=T_s}^{T_{s+1}-1} l_t(k_s) \\
&\le O\left(\frac{H \log T}{\eta} + T\eta\right) - \mathbb{E}\left[\frac{\rho_T(h^\dagger)}{40\eta \log T}\right] \\
&\quad + \alpha T(H-1) + \beta T(K-1) \\
&\quad + \mathbb{E}\left[2S \log(1/\beta)\sqrt{\frac{KT\rho_T(h^\dagger)}{h^\dagger}} + \frac{1}{2}\sqrt{SKT\rho_T(h^\dagger)}\right] \\
&= \tilde{O}\left(\mathbb{E}\left[\sqrt{SKT\rho_T(h^\dagger)}\right]\right) - \mathbb{E}\left[\frac{\rho_T(h^\dagger)\sqrt{TK}}{40\sqrt{H}\log(T)}\right],
\end{aligned}
\tag{49}
$$

where $\alpha = 1/(TH)$, $\beta = 1/(TK)$, $\eta = \sqrt{H/T}$, $H = \log(T)$, and $h^\dagger = \Theta(S)$. Then we can obtain

$$
R_S(T) = \tilde{O}\left(\min\left\{\sqrt{SKT\rho}, S\sqrt{KT}\right\}\right),
$$

where $\tilde{O}(S\sqrt{KT})$ is obtained from the worst case of $\rho_T(h^\dagger)$. The worst case can be found by considering a maximum value of the concave bound of the last equality in equation 49 with variable $\rho_T(h^\dagger) > 0$ such that $\rho_T(h^\dagger) = \tilde{\Theta}(S)$. This concludes the proof.

