# OpenReview forum: "Adversarial Bandits Against Arbitrary Strategies"
_TMLR — Accepted by TMLR_

### Review · Reviewer_6gDh · 2025-06-09

**Summary Of Contributions:**

The paper addresses the adversarial bandit problem where the learner’s performance is measured against the best “S-switched” sequence of arms in hindsight, for some switching parameter S. Existing results for this problem usually assumed S to be known, however the focus of the paper is for the setting where S is unknown. In this setup, two results are shown. Firstly, an algorithm based on the master-base framework and the online mirror descent (with fixed learning rate) is derived, and for which regret bounds are obtained. Secondly, an improved version of this algorithm is obtained (with adaptive learning rate) for which improved regret bounds w.r.t T are obtained.

**Audience:**

Yes

**Broader Impact Concerns:**

None.

**Claims And Evidence:**

Yes

**Requested Changes:**

Apart from addressing the points raised under "weaknesses" above, I think it might be helpful to provide a table upfront with an outline of the regret bounds for the switched setting. This would make it easier to compare the existing bounds with those in this paper.

**Strengths And Weaknesses:**

Strengths
----------------

1.	The paper is written well overall with a clear exposition, including: description of existing results, main ideas in the present work etc.

2.	The switched setting in bandit problems is notably challenging, and when S is unknown, the regret bound improves upon the results of (Luo et al. 2022). The idea of using the master-base framework along with the OMD method seems to exist in the literature, but using it for this particular problem is non-trivial. The work would be of interest for researchers working in online learning, as it provides improved regret bounds for adversarial bandits with switched arms.

Weaknesses
---------------------

1.	There are two methods derived, but it would be helpful to explicitly mention the regimes of S, K and T in which the regret bound of Alg. 1 is worse (or better?) than Alg. 2. It seems from the expressions of the regret bounds that no one method is strictly better in general. There is some discussion in this regard on page 8, but it seems to only discuss w.r.t T.

2.	In terms of the proof techniques themselves, it would be good to discuss clearly somewhere how it compares to existing proofs for these problems. This would also help the reader understand about the additional novelty in this respect.

---

> ### Author Response · Authors · 2025-06-21
>
> Thank you very much for your helpful and valuable comments. Please find our point-by-point responses below.
>
> **There are two methods derived, but it would be helpful to explicitly mention the regimes of S, K and T in which the regret bound of Alg. 1 is worse (or better?) than Alg. 2. It seems from the expressions of the regret bounds that no one method is strictly better in general. There is some discussion in this regard on page 8, but it seems to only discuss w.r.t T.**
>
> **Answer:** You are correct. Our algorithms are designed to perform well across different regimes, particularly with respect to $T$ and $S$. To recall the regret bounds, Algorithm1 achieves a regret of order $S^{1/2} K^{1/3} T^{2/3}$, while Algorithm2 achieves $\min\\{\sqrt{SKT\rho}, S\sqrt{KT}\\}$. This indicates that Algorithm 1 is advantageous when $S$ is large, whereas Algorithm 2 is preferable for larger $T$ due to its use of an adaptive learning rate that accounts for the variance of the loss estimator. We've added this discussion in Remark 3.5 in our revision.
>
> **In terms of the proof techniques themselves, it would be good to discuss clearly somewhere how it compares to existing proofs for these problems. This would also help the reader understand the additional novelty in this respect.**
>
> **Answer:** The main difference in the proof lies in how the master-base OMD framework is adapted to handle switching between $S$. First, the algorithms employ clipped probability simplices $\mathcal{P}\_{H}\^\alpha$ and $\mathcal{P}\_{H}\^\beta$, resulting in equations (17) and (18), which enhance the robustness of the algorithms with respect to the variance of the loss estimator and switching behavior, respectively. More importantly, Algorithm 2 employs an adaptive learning rate based on the variance, as supported by the theoretical insight provided in Lemma 3.3.
>
>
> **Apart from addressing the points raised under "weaknesses" above, I think it might be helpful to provide a table upfront with an outline of the regret bounds for the switched setting. This would make it easier to compare the existing bounds with those in this paper.**
>
>  **Answer:** Thank you for your suggestion. We have included a comparison table of the regret bounds (see Table 1 in Section 1) in our revision.

---

> > ### Author Response · Authors · 2025-06-22
> >
> > **Answer (Additional explanation regarding proof techniques):**
> > Here we provide additional explanation of the proof techniques used by our algorithms to handle switching between $S$. The algorithms employ clipped probability simplices $\mathcal{P}\_{H}\^\alpha$ and $\mathcal{P}\_{H}\^\beta$, resulting in equations (17) and (18), which enhance the robustness of the algorithms with respect to the variance of the loss estimator and the switching behavior, respectively. Specifically, the clipped domains guarantee that $p_{t,h^\dagger}(k) \ge \beta$ and $p_t(h^\dagger) \ge \alpha$, leading to bounds for the divergences in equations (17) and (18) as
> >
> > $$
> > D\_{F\_{\xi(h\^\dagger)}}(\mathbf{p}, \mathbf{p}\_{t\_s,h^\dagger})
> > \le \frac{\log(1/\beta)}{\xi(h\^\dagger)},
> > $$
> >
> > and
> >
> > $$
> > \sum_{t=t\_s}^{t\_{s+1}-1}
> > \mathbb{E}\big[
> > D\_{F\_{\xi(h^\dagger)}}(\mathbf{p}\_{t,h^\dagger}, \tilde{\mathbf{p}}\_{t+1,h^\dagger})
> > \big]
> > \le \frac{\xi(h^\dagger) K T\_s}{2\alpha}.
> > $$
> >
> > More importantly, Algorithm 2 employs an adaptive learning rate based on the variance, supported by the theoretical insight in Lemma 3.3, which yields a bound of $\sqrt{\rho\_T(h^\dagger)}$ for the regret with estimators. In particular,
> >
> > $$
> > \sum\_{t=t\_s}^{t\_{s+1}-1}
> > \mathbb{E}\big[ \langle \mathbf{p}\_{t,h\^\dagger} - \mathbf{p}, \mathbf{l}\_{t,h\^\dagger}\^{\prime\prime} \rangle \big] \le  \mathbb{E} \Big[ 2 \log(1/\beta) \sqrt{ \frac{K T \rho\_T(h\^\dagger)}{h\^\dagger} } + \frac{T\_s}{2} \sqrt{ \frac{S K \rho\_T(h\^\dagger)}{T} } \Big].
> > $$
> > This bound is critical for achieving $\sqrt{T}$ regret, even in the worst case, in conjunction with the log-barrier regularized master.

---

> > > ### Comment · Reviewer_6gDh · 2025-06-30
> > > **Read authors response**
> > >
> > > Thank you for answering my queries, I am fine with the explanations provided and have no further questions from my side.

---

### Review · Reviewer_TF1V · 2025-06-10

**Summary Of Contributions:**

The paper studies adversarial bandit against arbitrary strategies. Compared with the classical setting, the comparator policy can switch at most $S$ times, and the difficulty lies in that the parameter $S$ is unknown to the agent. To address this problem, this paper employs a master-based framework with the online mirror descent (OMD) algorithm. Two parallel results are provided: an $O(T^{2/3})$ regret with a fixed learning rate, and an $O(T^{1/2})$ regret with an adaptive one.

**Audience:**

Yes

**Broader Impact Concerns:**

The paper could serve as an interesting supplementary contribution to the bandit literature, particularly in the study of adversarial bandits. The author may consider adding a Broader Impact Statement to clarify the potential implications of the work.

**Claims And Evidence:**

Yes

**Requested Changes:**

1. The algorithm description and proof sketch part should be polished.

2. A detailed discussion on related works.

**Strengths And Weaknesses:**

Strengths:

Although I have not verified every detail of the proof, it appears sound and supports the claimed theoretical results.

Weaknesses:

1. I find the presentation somewhat unclear, partly due to inconsistent notation. For example, the learning rate $\eta$ is sometimes defined as a function of $h$, other times as a function of both $t$ and $h$, treated as a constant, or at times denoted by $\xi$. This inconsistency in notation makes it difficult to get the precise meaning in different parts of the paper. Moreover, the proof sketch jumps directly into the mathematical analysis without providing any intuitive explanation, which makes it less helpful for understanding.

2. The presentation of the algorithm lacks explanation. For example, in Section 3.4, why can $\rho_t$ be viewed as a variance term?  Why does the learning rate depend on $\rho_t$? Many of these algorithmic design choices appear to be based on Agarwal et al. (2017), but without clear explanations, readers unfamiliar with that work may find it difficult to follow. As a result, the presentation is not very self-contained. Another issue is in Lemma A.4, which directly refers to Lemma 13 in Agarwal et al. (2017) without providing a proof. It would be preferable to include a proof using the notation and framework established in this paper to maintain consistency and improve accessibility for readers.

3. The proposed algorithm seems quite relevant to Agarwal et al. (2017) and Luo et al. (2022). To me, I feel that it’s a direct application of the master algorithm proposed in Agarwal et al. (2017) to various base-algorithms with different hyperparameters $h$, as a conjecture of the unknown $S$. I believe the paper would benefit from a more thorough discussion and a detailed comparison with these works.

4. (Some minor issues) equation (13): There should not be a summation over $S$. Lemma 3.3 is exactly the same as Lemma A.3.

---

> ### Author Response · Authors · 2025-06-21
>
> Thank you very much for your helpful and constructive comments. Please find our point-by-point responses below.
>
> **The presentation is sometimes unclear due to inconsistent notation for learning rates.**
>
> **Answer:** Thank you for your valuable suggestions. In our revision, you can find that we have clarified the notation for the learning rates of the master and the bases.  Specifically, for the basic master-base OMD algorithm (Section 3.3), we denote the master’s learning rate in the regularizer by $\eta$, and the learning rate for base $h$ by $\xi(h)$.  Similarly, for the master-base OMD with adaptive learning rates (Section 3.4), we use $\eta_t(h)$ to represent the master’s adaptive learning rate for handling base $h$ at time $t$, and $\xi_t(h)$ for the adaptive learning rate of base $h$ at time $t$.
>
> **The proof sketch jumps directly into the mathematical analysis without providing any intuitive explanation.**
>
> **Answer:** For the proof, we have included an intuitive explanation in the revised proof sketch, illustrating how our regret analysis is decomposed into the regret from the master and from each base, which are detailed in the revised sketches for Theorems 3.1 and 3.2, highlighted in red.
>
> **In Section 3.4, why can $\rho_t$ be viewed as a variance term? Why does the learning rate depend on $\rho_t$?**
>
> **Answer:**  Thank you for your valuable suggestions. First, we explain why $\rho_t(h)$ can be interpreted as a variance term. In Algorithm 2, $\rho_t(h)$ is updated at every doubling of the inverse probability $1/p_t(h)$, where $p_t(h)$ is the probability of selecting base $h$ by the master. The loss estimator for each base is defined as $l_t'(h) = \frac{l_t(a_t) \mathbf{1}(h_t = h)}{p_t(h)},$
> which has variance on the order of $ \mathrm{Var}[l_t'(h)] \approx \frac{p_t(h)}{p_t(h)^2} = \frac{1}{p_t(h)}.$ Moreover, when considering the loss estimator for each arm $a$ chosen by base $h$, $l_{t,h}''(a) = \frac{l_t'(h)}{p_{t,h}(a)},$ the variance increases accordingly.
>
> Since the estimator's variance depends inversely on the selection probabilities, the basic OMD method described in Section 3.3 achieves a regret bound of order $O(T^{2/3})$ by using a learning rate independent of variance. However, by incorporating $\rho_t(h)$ into the adaptive learning rate $\xi_t(h) = \sqrt{\frac{h}{K T \rho_t(h)}},$ we can optimize the regret bound to depend on $\sqrt{\rho_T(h^\dagger)}$, as demonstrated in Lemma A.3, and achieve $O(\sqrt{T})$ in Section 3.4.
>
>
> **The proposed algorithm seems quite relevant to Agarwal et al. (2017) and Luo et al. (2022).**
>
> **Answer:** It is true that we adopt the master--base framework to handle the unknown $S$, using the algorithm from Agarwal et al.(2017) as the master. However, a straightforward application does not guarantee the $O(\sqrt{T})$ regret bound achieved in Section3.4.
>
> Since we utilize the estimator, which variance depends inversely on the selection probabilities, by incorporating $\rho_t(h)$ into the adaptive learning rate for each base $h$, $\xi_t(h) = \sqrt{\frac{h}{K T \rho_t(h)}},$ we can optimize the regret bound to depend on $\sqrt{\rho_T(h^\dagger)}$, as demonstrated in Lemma 3.3, and achieve $O(\sqrt{T})$ in Section 3.4.  Notably, this adaptive base algorithm is effectively combined with the master employing log-barrier regularization [Agarwal et al. (2017)] to control the regret due to variance—an integration that, to the best of our knowledge, has not been explored before (eq. 13). This novel integration is the main reason why Algorithm 2 can achieve $\sqrt{T}$, even in the worst case, without the knowledge of $S$ (Remark 3.4).
>
> **Another issue is in Lemma A.4, which directly refers to Lemma 13 in Agarwal et al. (2017) without providing a proof.**
>
> **Answer:**  Thank you for your suggestions. Lemma A.4, as we mentioned, can be readily derived from Lemma 13 in Agarwal et al. (2017), except for the adjustment due to the clipped probability domain. To improve accessibility, we have included a brief guiding proof for the lemma, highlighted in red in our revision. We are happy to provide additional details if they are necessary.
>
> **The algorithm description and proof sketch part should be polished.**
>
> **Answer:** In our revision, we have provided a more detailed description of the OMD framework to help readers better understand the approach adopted (Section 3.2). We have also revised the notations for the learning rates in Sections 3.3 and clarified the role of the variance term in defining the adaptive learning rate.  The revised proof sketch with a structured description can be found in the updated manuscript.
>
> **A detailed discussion on related works.**
>
> **Answer:** We have included a comparison table (Table 1 in Section 1) summarizing previous algorithms alongside our proposed methods. Furthermore, we have expanded the discussion of the novelty of our approach compared to related work (Remark 3.4).

---

> > ### Author Response · Authors · 2025-06-22
> >
> > **(Some minor issues) equation (13): There should not be a summation over $\mathcal{S}$. Lemma 3.3 is exactly the same as Lemma A.3.**
> >
> > **Answer:** Thank you for your valuable feedback. We have corrected equation (13). In the previous version, to prove Lemma 3.3, we reintroduced Lemma A.3 in the appendix with its proof. We have revised this part by indicating the restatement to eliminate any confusion.

---

### Review · Reviewer_zzRq · 2025-07-20

**Summary Of Contributions:**

This work focus on the adversarial bandit and design an algorithm whose regret depends on the number of best-arm switches.

**Audience:**

Yes

**Claims And Evidence:**

Yes

**Requested Changes:**

Please see the 'Con' part in the previous session.

**Strengths And Weaknesses:**

Pro:
1. The paper roughly looks good and has some new result.
2. Comparison to some existing paper is discussed.

Con:
1. What is the analytical challenge here? Please highlight it.
1. Adversarial bandit is an interesting topic, while still, further emphasis of the contribution of this work is appreciated.
1. how far is the derived upper bound away from the lower bound?
1. Is an adaptive adversary considered here?
1. What is the value of $rho$? How does it depend on $S$? Some discussion on value of $rho$, at least in special case would benefit the implication of superiority of the new result. Otherwise, it is not clear whether the proposed algorithm is better.
1.  Numerial experiments are appreciated.

---

> ### Author Response · Authors · 2025-07-24
>
> Thank you very much for your helpful and constructive comments. Please find our point-by-point responses.
>
> **What is the analytical challenge here? Please highlight it.**
>
>  **Answer:**
>    The key analytical challenge lies in designing a hierarchical Online Mirror Descent (OMD) framework that remains robust under an unknown number of switches $S$ while controlling the estimator's variance.
>
> To address this, we introduce *clipped probability simplices*, $\mathcal{P}\_{H}\^\alpha$ and $\mathcal{P}\_{H}\^\beta$, which ensure the probabilities are lower-bounded by $\alpha$ and $\beta$, respectively. This clipping enables control over the divergences arising in our analysis, leading to the following bounds:
>
> $ D\_{F\_{\xi(h^\dagger)}}(\mathbf{p}, \mathbf{p}\_{t\_s,h^\dagger})
>     \le \frac{\log(1/\beta)}{\xi(h^\dagger)}$ and
>     $\sum\_{t=t\_s}^{t\_{s+1}-1}
>     \mathbb{E}\left[
>     D\_{F\_{\xi(h\^\dagger)}}(\mathbf{p}\_{t,h^\dagger}, \tilde{\mathbf{p}}\_{t+1,h^\dagger})
>     \right]
>     \le \frac{\xi(h^\dagger) K T\_s}{2\alpha}.$  (Eqs. (17) and (18))
>
> In addition, Algorithm 2 leverages an *adaptive learning rate* based on the variance of the loss estimators. Lemma 3.3 supports this design by bounding the regret against any comparator $\mathbf{p}$:
>
>    $$ \sum\_{t=t\_s}^{t\_{s+1}-1}
>     \mathbb{E}\left[
>     \langle \mathbf{p}\_{t,h^\dagger} - \mathbf{p},\,
>     \mathbf{l}\_{t,h^\dagger}^{\prime\prime} \rangle
>     \right]
>     \le
>     \mathbb{E} \left[
>     2 \log(1/\beta)
>     \sqrt{ \frac{K T \rho\_T(h^\dagger)}{h^\dagger} }
>     + \frac{T\_s}{2}
>     \sqrt{ \frac{S K \rho\_T(h^\dagger)}{T} }
>     \right].$$
>
> This variance-sensitive bound is crucial in achieving $\widetilde{O}(\sqrt{T})$ regret even under worst-case switching behavior, when combined with the log-barrier regularized master algorithm.
>
> **Adversarial bandit is an interesting topic, while still, further emphasis of the contribution of this work is appreciated.**
>
>   **Answer:** We appreciate the reviewer’s interest. Our contribution goes beyond the classical adversarial bandit setting, which typically competes against a fixed best arm. Instead, we consider a more general and realistic scenario: the learner competes against a sequence of best arms with an unknown number of switches $S$, significantly complicating the problem.
>
> Our main contributions are:
> - Propose a hierarchical OMD framework tailored to switching environments;
> - The use of clipped probability domains and adaptive learning rates to manage estimator variance and switching complexity;
> - New regret bounds for the switching regret under unknown $S$, improving upon prior approaches such as BOB (Cheung et al., 2019) in terms of the dependence on the time horizon $T$.
>
> These elements form a cohesive and technically novel solution to a more challenging version of the adversarial bandit problem.
>
>
> **how far is the derived upper bound away from the lower bound?**
>
> **Answer:**
> In the case where $S$ is known, the minimax regret lower bound is $\Omega(\sqrt{SKT})$. Our setting is more challenging as it assumes unknown $S$.
>
> For this setting of unknown $S$, the minimax lower bound remains an open problem. Nevertheless, we provide regret guarantees of $\widetilde{O}(S^{1/2}K^{2/3}T^{2/3})$ and $\widetilde{O}(\min\\{\sqrt{SKT\rho}, S\sqrt{KT}\\})$, which represent improvements over prior work, BOB (Cheung et al., 2019), which achieves only $\widetilde{O}(T^{3/4})$ under the same assumptions. We believe our results offer new insights and serve as a stepping stone toward closing the gap between achievable upper and potential lower bounds in the unknown-$S$ setting.
>
> **Is an adaptive adversary considered here?**
>
> **Answer**: Yes, our theoretical results are valid against adaptive adversaries that may select losses based on the learner’s past actions. This is consistent with the standard setting in the adversarial bandit literature.
>
>
> **What is the value of $\rho$? Discussion at least in special case would benefit the implication of superiority of the new result.**
>
> **Answer:**
> Thank you for raising this point. The quantity $\rho$ stems from the variance of the importance-weighted loss estimator in our algorithm. Its precise value depends on the interaction between the loss sequence and the learner’s sampling distribution and can vary across instances.
>
> Instead of assuming a fixed value or distribution for $\rho$, our analysis yields worst-case regret bounds that hold for all $\rho$ such that Algorithm 2 achieves $\tilde{O}(\sqrt{T})$ regret in Theorem 3.2, while  Algorithm 1 achieves $\widetilde{O}(T^{2/3})$, and prior work (BOB) achieves only $\widetilde{O}(T^{3/4})$ in the same setting.
>
> **Numerical experiment**
>
> **Answer:**
> In line with most prior work in adversarial bandits, our focus in this paper is primarily theoretical. We believe establishing solid theoretical foundations is essential in this challenging setting. We view empirical validation as a valuable direction for future work.

---

### Decision · Action_Editor_wguF · 2025-09-29

**Recommendation:** Accept with minor revision

**Additional Comments:**

The reviewers found the obtained regret bounds interesting and useful but expressed concerns on novelty and presentation. The revision added some helpful explanations, including more explicit comparison against prior works.  Reviewer zzRq also suggested experimental validation, which the authors did not consider necessary as the main results are purely theoretical.

Overall, the contribution seems moderate: the algorithm design and proof techniques were adapted from prior work to the setting of unknown number of switches in the best arm. There are some typos and inaccuracies that may hurt the credibility of the results (but none seemed unfixable perhaps). Please see below for a list of issues that we spot:

- Section 2: It is defined A = [K] at the beginning. Yet, the notation $A^T$ and $[K]^T$ both appeared for no reason.

- Section 3.2: equation 3.2 cannot be found (presumably the second displayed equation in this section).

- Section 3.3: F_{t-1} is the filtration. Filtration generated by what?

- Statement of Theorem 3.1: from the proof it is clear that there is an additive factor of $\sqrt{KT}$, see page 6 about the term $\eta TK/2$, where $\eta = 1/\sqrt{TK}$. Yet, the theorem claims $K^{1/3}$ dependence. In the response to Reviewer zzRq, the authors suddenly switched to $K^{2/3}$. Please fix this in the final revision.

- Section 3.4: $F_{\xi}$ and $F_{\eta}$ are used to denote two different Legendre functions... Better use different letters.

- Remark 3.6: the claim that mixing with uniform is enough to solve the projection onto the clipped simplex needs justification.

- Lemma A.2: p_1 is defined as uniform, which is not the minimizer of F over the clipped simplex (an assumption to invoke Lattimore & Szepesvari). This is minor if one makes assumption on how large K or alpha is, but this discussion should be included.

- Equation (23): no summation over s.

- There are some minor typos in Eq (26): should be $D_F(p, p_{t,h^\dagger})$ and $\xi_{t_s}$ on line 3 and line 5, respectively.

- Above Eq (28): should be equation 23 (instead of 20)?

- Lemma A.4: Why not include the entire proof to make it self-contained?

It is recommended that the authors go through the entire paper carefully and fix any remaining typos before uploading the final revision.

**Audience:**

Yes

**Audience Explanation:**

The bandit problem is widely studied in online learning and has many applications in machine learning.

**Claims And Evidence:**

Yes

**Claims Explanation:**

The claims are backed up by proofs. However, there are some typos and inaccuracies that need to be fixed.

---

> ### Author Response · Authors · 2025-10-17
>
> We sincerely thank the Action Editor and reviewers for the careful reading of our paper and for the constructive and detailed feedback.
> We have carefully addressed all the suggested revisions and corrected typos, notational inconsistencies, and missing details accordingly.
> Below we summarize the modifications made in response to each comment.
>
>  - Section 2: It is defined $A = [K]$ at the beginning. Yet, the notation $A^T$ and $[K]^T$ both appeared for no reason.
>
>  **Response:** We have unified the notation and now consistently use $[K]$ to denote the action set $\mathcal{A}$ throughout the paper.
>
>   - Section 3.2: Equation 3.2 cannot be found (presumably the second displayed equation in this section).
>
>   **Response:** We have explicitly labeled the OMD update equation as (1) and updated all corresponding references accordingly.
>
>   - Section 3.3: $\mathcal{F}_{t-1}$ is the filtration. Filtration generated by what?
>
>   **Response:** We have added a clarification that $\mathcal{F}_{t}$ denotes the natural filtration generated by the history up to round $t$.
>
>   - Statement of Theorem 3.1: From the proof it is clear that there is an additive factor of $\sqrt{KT}$, see page6 about the term $\eta TK / 2$, where $\eta = 1/\sqrt{TK}$. Yet, the theorem claims $K^{1/3}$ dependence. In the response to Reviewer zzRq, the authors suddenly switched to $K^{2/3}$. Please fix this in the final revision.
>
>  **Response:** We have corrected the proof to include the factor $\sqrt{TH} = \tilde{O}(\sqrt{T})$ instead of $\sqrt{KT}$.
>
> - Section 3.4: $F_{\xi}$ and $F_{\eta}$ are used to denote two different Legendre functions. Better use different letters.
>
> **Response:** We now use distinct symbols $\mathcal{F}$ and $\mathcal{G}$ for the two Legendre functions to avoid confusion.
>
>  - Remark 3.6: The claim that mixing with uniform is enough to solve the projection onto the clipped simplex needs justification.
>
> **Response:** We have clarified the remark regarding the implementation.
>
>  - Lemma A.2: $p_1$ is defined as uniform, which is not the minimizer of $F$ over the clipped simplex (an assumption to invoke Lattimore \& Szepesvári). This is minor if one makes assumption on how large $K$ or $\alpha$ is, but this discussion should be included.
>
> **Response:** We have added a detailed proof for Lemma~A.2 and adjusted the lemma  to account for uniform initialization.
>
> - Equation (23): No summation over $s$.
>
> **Response:** We have corrected the equation by explicitly including the summation over $s$ in Equation (34) (in the revised version).
>
> - Equation (26): There are some minor typos: it should be $D_F(p, p_{t,h^\dagger})$ and $\xi_{t_s}$ on line 3 and line 5, respectively.
>
>  **Response:** We have corrected some notations in Equation (37) (in the revised version).
>
>  - Above Equation (28): Should be equation (23) (instead of~20)?
>
>  **Response:** We have fixed the incorrect reference, replacing it with “Equation (34)” in the sentence preceding Equation (39) (in the revised version).
>
>  - Lemma A.4: Why not include the entire proof to make it self-contained?
>
>  **Response:** We have included the full proof of Lemma A.4 in the appendix to make the presentation self-contained.